# ReSched: Rethinking Flexible Job Shop Scheduling from a Transformer-based Architecture with Simplified States

**Xiangjie Xiao**[1], **Cong Zhang**[2], **Wen Song**[3], **Zhiguang Cao**[1]*

[1]School of Computing and Information Systems, Singapore Management University, Singapore
[2]College of Computing and Data Science, Nanyang Technological University, Singapore
[3]Institute of Marine Science and Technology, Shandong University, China
`xj.xiao.2024@phdcs.smu.edu.sg, cong.zhang92@gmail.com,`
`wensong@email.sdu.edu.cn, zgcao@smu.edu.sg`

## ABSTRACT

Neural approaches to the Flexible Job Shop Scheduling Problem (FJSP), particularly those based on deep reinforcement learning (DRL), have gained growing attention in recent years. However, existing methods rely on complex feature-engineered state representations (i.e., often requiring more than 20 handcrafted features) and graph-biased neural architectures. To reduce modeling complexity and advance a more generalizable framework for FJSP, we introduce ReSched, a minimalist DRL framework that rethinks both the scheduling formulation and model design. First, by revisiting the Markov Decision Process (MDP) formulation of FJSP, we condense the state space to just four essential features, eliminating historical dependencies through a subproblem-based perspective. Second, we employ Transformer blocks with dot-product attention, augmented by three lightweight but effective architectural modifications tailored to scheduling tasks. Extensive experiments show that ReSched outperforms classical dispatching rules and state-of-the-art DRL methods on FJSP. Moreover, ReSched also generalizes well to the Job Shop Scheduling Problem (JSSP) and the Flexible Flow Shop Scheduling Problem (FFSP), achieving competitive performance against neural baselines specifically designed for these variants.

## 1 INTRODUCTION

The Flexible Job Shop Scheduling Problem (FJSP) is a fundamental combinatorial optimization problem (COP) with widespread applications in manufacturing (Ding et al., 2019; Wang et al., 2024a), edge computing (Luo et al., 2021; Yang et al., 2025), and logistics (SAT, 2014; Arunarani et al., 2019). In FJSP, jobs are decomposed into sequences of operations, each processable by a set of compatible machines. Solving FJSP entails two coupled decisions: assigning operations to machines and sequencing them on each machine. As a generic model, FJSP unifies multiple scheduling scenarios: it reduces to the Job Shop Scheduling Problem (JSSP) when machine assignments are fixed, and to the Flexible Flow Shop Scheduling Problem (FFSP) when jobs follow a shared processing stage but retain machine flexibility. This inherent generality makes FJSP a critical framework for diverse real-world systems.

Recent research has leveraged Deep Reinforcement Learning (DRL) to construct scheduling heuristics (Feng et al., 2021; Lei et al., 2022), typically representing partial solutions (states) via disjunctive graphs (Błażewicz et al., 2000) enriched with complex node features. However, many of these features are redundant[1], and incorporating historical construction data into the current state can paradoxically degrade learning (see Section 4.1.2). Furthermore, pruning unpromising actions based on human heuristics (Song et al., 2023; Wang et al., 2024b; Zhao et al., 2025), while intended to boost

---

*Corresponding author.

[1]For instance, we demonstrate in Appendix B.3 using DANIEL (Wang et al., 2024b) that removing half of the input features does not compromise performance.

efficiency, often hampers policy generalization and yields suboptimal solutions (see Appendix B.3). These practices necessitate persistent tracking of auxiliary variables at every step, introducing significant computational overhead. Architecturally, most DRL approaches rely on Graph Attention Networks (GATs) (Velickovic et al., 2017), which impose rigid inductive biases: capturing long-range dependencies requires deep stacking, while linear attention mechanisms struggle to model complex non-local scheduling interactions. These limitations prompt a central question: *Can we design a general construction policy, derived from a minimal Markov-sufficient state and implemented with a generic yet expressive architecture, that generalizes naturally to FJSP variants?*

We address this by proposing RESCHED, a DRL framework that unifies minimalist state design with flexible neural modeling to achieve state-of-the-art (SOTA) performance. We demonstrate that ensuring state sufficiency reduces the need for heavy architectural inductive bias. Our formulation introduces a compact state representation comprising only four core node features and a graph structure that explicitly encodes intra-job dependencies, thereby eliminating the need for historical tracking. To enhance representational power, we replace conventional GNN-based policies (Franco et al., 2009) with a Transformer backbone featuring two complementary branches: self-attention for operations and cross-attention for machines. Applying Transformers to FJSP presents specific challenges: self-attention must capture process constraints without extra parameters, while cross-attention must handle indirect edge-feature integration and severe operation-machine imbalance (often exceeding 10:1). We overcome these by incorporating Rotary Positional Encoding (RoPE) for intra-job sequencing and determining a novel cross-attention module that directly embeds edge features while using self-connections to mitigate representation dilution. Empirical results show that RESCHED establishes a new SOTA on FJSP, outperforming both handcrafted heuristics and leading DRL baselines. Crucially, it generalizes robustly across problem sizes, datasets, and variants (JSSP, FFSP), proving that a minimalist design coupled with expressive modeling enables broad applicability. Our key contributions are:

- We revisit the MDP for FJSP to design a compact state representation with only four essential node features. By utilizing a graph structure that explicitly encodes intra-job relationships, we eliminate redundant features, historical dependencies, and auxiliary variables.

- We introduce a dual-branch Transformer architecture tailored for scheduling. It features self-attention enhanced by RoPE to model intra-job dependencies, and a novel cross-attention module that directly integrates edge features while mitigating operation–machine imbalance through self-connections.

- We demonstrate that RESCHED achieves SOTA performance on FJSP benchmarks while exhibiting strong generalization capabilities across varying problem sizes and scheduling variants, including JSSP and FFSP.

## 2    RELATED WORK

Priority dispatching rules (PDRs) (Haupt, 1989; Sels et al., 2012) are widely used in real-world FJSPs for their simplicity, interpretability, and fast decision-making. However, designing effective and generalizable PDRs remains challenging, as they often rely on domain expertise and fail to adapt across diverse problem instances. This limitation has motivated a surge of interest in learning PDRs through deep reinforcement learning (DRL). Most DRL-based methods formulate FJSP as a disjunctive graph, where nodes represent operations and machines, and edges encode precedence and assignment constraints. Graph neural networks (GNNs) are then employed to capture the complex relationships among operations and machines. For example, Song et al. (2023) proposed HGNN, a heterogeneous GNN tailored for FJSP; Wang et al. (2024b) introduced a dual-attention mechanism to jointly model operation and machine features; and Zhao et al. (2025) developed a GNN-based approach augmented with reward shaping to improve training efficiency. While effective, these methods typically depend on heavily engineered state representations, which limit scalability and generalization. Beyond FJSP, DRL has also been applied to other scheduling variants. For instance, Zhang et al. (2020) proposed a GNN-based DRL framework for JSSP that learns operation sequencing policies, and Kwon et al. (2021) introduced a mixed-score attention mechanism to model operation–machine interactions in FFSP. These works further highlight the growing role of neural methods in advancing data-driven scheduling.

## 3 PRELIMINARY

### 3.1 FLEXIBLE JOB SHOP SCHEDULING PROBLEM (FJSP)

Consider a generic scheduling problem with *operations* as fundamental units. Suppose there are two sets: a set of operations $\mathcal{O} = \{O_1, O_2, \ldots, O_n\}$ and a set of machines $\mathcal{M} = \{M_1, M_2, \ldots, M_m\}$. Each operation $O_i$ can be processed by one of the machines in $\mathcal{M}$ with a specific duration (processing time) $D_i^m > 0$. In FJSP, jobs are collections of operations that must be executed sequentially. Each job $J_i$ consists of several operations, so operations can be represented as $O_{ij}$, where $i$ denotes the job index and $j$ represents the position of the operation in job $i$. Each operation $O_{ij}$ can be processed by one or multiple machines, which means "flexible", from a compatible machine set $\mathcal{M}_{ij} \subseteq \mathcal{M}$ with duration $D_{ij}^m > 0$. Meanwhile, the FJSP is subject to several important *constraints*: (1) An operation can only begin after all its preceding operations in the job sequence have been completed. (2) Each operation must be assigned to a single eligible machine and executed non-preemptively, without interruption once started. (3) Each machine processes only one operation at a time, requiring sequential execution without overlap. The *objective* of FJSP is to find a feasible solution that satisfies all the above constraints while minimizing the overall makespan, which is defined as the maximum finish time among all operations. Moreover, the transformation of FJSP to other variants like JSSP and FFSP are presented in Appendix A.2.

### 3.2 TRANSFORMER BLOCK

The Transformer block (Vaswani et al., 2017) is a fundamental component designed to capture dependencies in sequential data. It primarily consists of a multi-head attention (MHA) mechanism and a feed-forward network (FFN), combined with residual connections and layer normalization (He et al., 2016; Ba et al., 2016). In the attention mechanism, each element attends to all others through a weighted combination of their representations. For a given input sequence $h = [h_1, h_2, \ldots, h_n]$, the attention weights are computed by measuring the similarity between a query $q_a \in \mathbb{R}^d$ and keys $k_b \in \mathbb{R}^d$, typically obtained via learned linear projections of $h_a$ and $h_b$. The normalized attention weight $\alpha_{a,b}$ from node $a$ to node $b$, and the resulting embedding $h'_a$ for node $a$ are computed as follows (head is omitted for brevity):

$$\alpha_{a,b} = \text{softmax}_b \left( \frac{\langle q_a, k_b \rangle}{\sqrt{d}} \right), \quad h'_a = \sum_{b=1}^{n} \alpha_{a,b} v_b, \tag{1}$$

where $\langle \cdot, \cdot \rangle$ denotes the dot product, $\sqrt{d}$ is a scaling factor to stabilize training, and $v_b \in \mathbb{R}^d$ is the value vector corresponding to node $b$. The outputs from multiple heads are concatenated and passed through a linear transformation to form the final MHA output, which is then processed by a position-wise FFN, with residual connections and layer normalization applied after both the attention and FFN sub-layers. This structure allows the Transformer block to effectively capture global dependencies and complex interactions among elements in the input.

## 4 METHODOLOGY

In this section, we introduce *ReSched*, a construction-type neural framework designed for solving scheduling problems. We begin by revisiting the problem formulation, focusing on the Flexible Job Shop Scheduling Problem (FJSP), a generalized model that captures diverse scheduling scenarios. Within this formulation, we cast scheduling as a Markov Decision Process (MDP), where each decision step resolves a subproblem using a compact, task-specific state representation. Building on this MDP, we develop a Transformer-based policy network equipped with structure-aware attention mechanisms and trained via reinforcement learning.

### 4.1 STATE REPRESENTATION

In our framework, we aim to minimize the complexity of the state representation while ensuring it remains fully expressive and sufficient to guide optimal scheduling decisions.

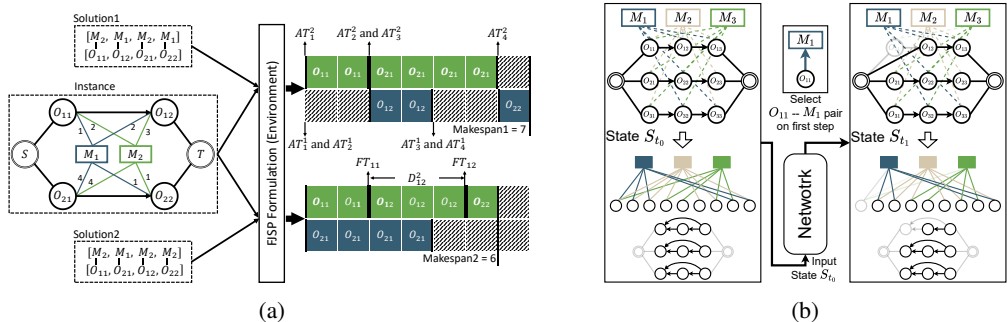

Figure 1: (a) Illustration of the formulation for a 2-job, 2-operation, 2-machine (2-2-2) FJSP instance. (b) Changes in topology and O2M/O2O Connection between two steps for a 3-3-3 instance.

### 4.1.1 REVISITING THE SCHEDULING FORMULATION

We represent an FJSP instance using a heterogeneous disjunctive graph (Song et al., 2023), as illustrated in Figure 1a. A solution to FJSP consists of two key components: *assignment* and *order*. To evaluate solution quality, an explicit formulation is required to compute the makespan.

Let $a_{t,ij}^m \in \{0, 1\}$ indicate assignment of $O_{ij}$ to machine $m$ at step $t$. To ensure only one operation-machine pair is scheduled per step, we enforce a constraint: $\sum_{m \in \mathcal{M}} \sum_{(ij) \in \mathcal{O}} a_{t,ij}^m = 1, \forall t$. When $a_{t,ij}^m = 1$, the finish time $FT_{ij}$ is computed as:

$$FT_{ij} = \max\left(FT_{i(j-1)}, AT_t^m\right) + D_{ij}^m, \quad \text{if } a_{t,ij}^m = 1,$$
$$AT_t^m = \begin{cases} FT_{i'j'} & \text{if } a_{t-1,i'j'}^m = 1 \\ AT_{t-1}^m & \text{otherwise.} \end{cases} \quad (2)$$

Here, $FT_{i(j-1)}$ denotes the finish time of the preceding operation in job $J_i$, $AT_t^m$ is the available time of machine $m$ at step $t$, and $(i'j')$ is the operation assigned to machine $m$ at the previous step. The goal of FJSP is to minimize the maximum finish time across all operations, which is defined as $FT_{\max} = \max_{(ij) \in \mathcal{O}} FT_{ij}$. This unified formulation supports our state representation simplification, and it can also be adapted to JSSP and FFSP by appropriately modifying machine assignment rules. For the sake of space, the detailed mathematical formulation of FJSP and its variants are deferred to Appendix A.1.

Looking at Figure 1a again, it illustrates how two different solutions for the same FJSP instance are translated, via the above formulation, into concrete Gantt charts and makespan values. Specifically, the top solution with order $[O_{11}, O_{12}, O_{21}, O_{22}]$ and assignment $[M_2, M_1, M_2, M_1]$ results in a makespan of $FT_{\max} = 7$, while the bottom solution with order $[O_{11}, O_{21}, O_{12}, O_{22}]$ and assignment $[M_2, M_1, M_2, M_2]$ achieves a makespan of $FT_{\max} = 6$.

### 4.1.2 MDP FORMULATION

As a construction method, the scheduling process can be viewed as a sequential decision-making problem, where the agent iteratively selects an operation-machine pair to assign at each step. This leads to a natural formulation of the scheduling problem as an MDP.

**State: Minimal Representation** According to Eq. (2), computing the finish time for operation $O_{ij}$ at step $t$ requires three pieces of information: (1) the finish time of its predecessor $O_{i(j-1)}$, (2) the *Duration $D_{ij}^m$* of $O_{ij}$ on machine $M_m$, and (3) the available time $AT_t^m$ of machine $M_m$. Particularly, "predecessor" denotes the precedence constraint between operations: an operation may start only after all of its predecessors have finished; we refer to this as the *operation-to-operation (O2O) dependency*. Meanwhile, the *Machine Available Time $AT_t^m$* is determined by the finish time of the operation assigned to machine $M_m$ at the previous steps. Therefore, the $AT_t^m$ is influenced by the *operation-to-machine (O2M) connection*. From a consistency perspective, the finish time of $O_{i(j-1)}$ can also be interpreted as the *Operation Available Time* for its successor $O_{ij}$.

Thus, the transition from step $t-1$ to $t$ requires only the following information: 1) Operation and Machine Available Time (Node feature); 2) Duration (Edge feature); 3) O2O Dependency (Graph structure); 4) O2M Connection (Graph structure).

**Definition 4.1.** *Let $\mathcal{S}_t \in \mathbb{S}$ be the state representation at decision step $t$, which uniquely determines the following: 1) the available times of all machines, 2) the completion status of all jobs (i.e., the finish times of their most recently scheduled operations), 3) the O2O precedence between operations, and 4) the O2M connections (including the corresponding durations).*

**Proposition 1** (State-dependent Optimality in Scheduling). *For any two scheduling trajectories $\tau_1$ and $\tau_2$ that reach the same state $\mathcal{S}_t$, the corresponding remaining subproblems share an identical feasible solution set.*

As a direct consequence, the optimal decision at step $t$ depends only on $\mathcal{S}_t$, rather than on the full trajectory history. With this state definition, the scheduling problem can be viewed as a finite-state MDP that satisfies the Markov property. A formal proof of Proposition 1 is provided in Appendix D.

**State: Subproblem**    In our framework, each scheduling step is modeled as an individual subproblem. To better support this formulation, we refine the state representation along two dimensions: available time from *node features* and O2O dependencies from *graph structure*, thereby excluding historical information and focusing solely on the current subproblem. 1) Relative Available Time. We normalize all operation and machine available times by subtracting the global minimum available time at each step. This prevents the unbounded growth of absolute time values and mitigates potential generalization issues. A conceptually related idea of removing historical dependencies and using relative-time features for JSSP was explored by Lee & Kim (2024); Unlike this JSSP-specific design, we remove all historical information under a subproblem-based formulation and extend the relative-time principle to a unified minimal state for FJSP and its variants. 2) O2O Connection. Instead of the bidirectional operation-to-operation (O2O) edges commonly used in prior work (Song et al., 2023; Wang et al., 2024b; Zhao et al., 2025), we adopt backward-looking edges. Under the subproblem formulation, each operation only requires information from its successors, making redundant historical tracking unnecessary. To further improve efficiency, we introduce hop connections from each operation to all its successors, granting direct access to job-level future constraints without relying on multi-layer message passing. Figure 1b illustrates how graph topology and O2O/O2M connections evolve: once an operation is scheduled, it and its associated O2O/O2M connections are removed, yielding a new subproblem.

Regarding the node feature mentioned in *Minimal Representation*, we also incorporate the Minimum Duration $\min_{m \in \mathcal{M}_{ij}} D_{ij}^m$ across candidate machines as a compact yet informative proxy. This value provides a lower bound on processing time, enabling the network to distinguish operations of varying difficulty. Although it depends on the set of candidate machines and may be sensitive to instance variations, prior work (Song et al., 2023; Lei et al., 2022; Wang et al., 2024b; Zhao et al., 2025) shows that it is a simple and effective approximation in practice. In our framework, it significantly improves learning efficiency without requiring explicit machine assignments.

**State: Features**    Our state representation consists of four key features: 1) Operation Available Time; 2) Machine Available Time; 3) Duration; and 4) Minimum Duration. Note that dependency and machine eligibility are not features but graph structure, represented by O2O/O2M connections.

**Action**    In our MDP formulation, the action at step $t$ corresponds to selecting an operation–machine pair $(ij, m)$, meaning that operation $O_{ij}$ is assigned to machine $M_m$. For simplicity, we denote this action as $a_t$, and the complete schedule is represented as a sequence of actions: $\mathcal{A} = \{a_1, \cdots, a_t, \cdots, a_n\}$, where $n$ is the total number of scheduling steps (i.e., the total number of operations). In contrast to many existing neural approaches for FJSP, which introduce auxiliary notions such as free time or current time to prune the action space at each step (Song et al., 2023; Wang et al., 2024b; Zhao et al., 2025), our framework avoids such heuristic constraints. The only restriction we impose is the natural precedence constraint between operations (Zhang et al., 2020).

**Transition**    After taking action $a_t$, the environment transitions deterministically to a new state $s_{t+1}$, fully determined by the current state $s_t$ and action $a_t$. Specifically, operation and machine status are updated according to Eq. (2).

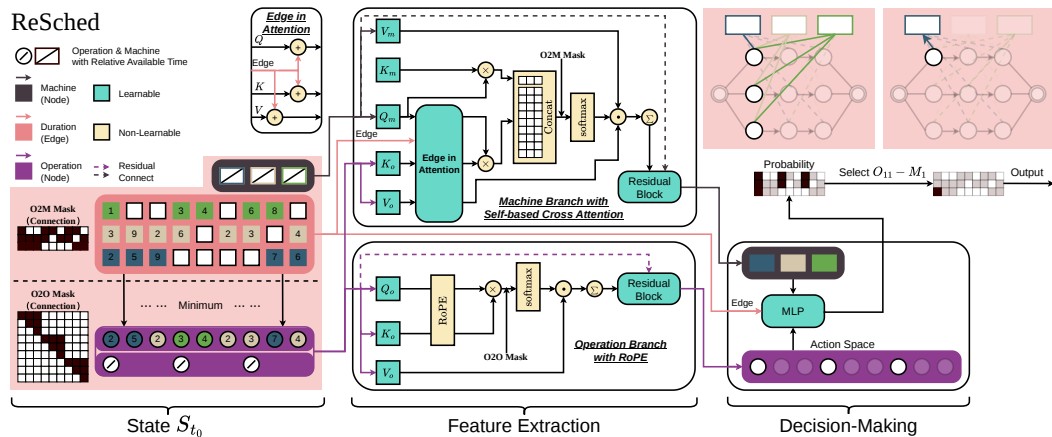

Figure 2: The RESCHED framework. The state consists of four information, resulting in four key features (available time and minimum duration for operation; available time for machine; duration), and incorporates three (underlined) architectural enhancements for Transformer-based network. For decision-making module, we concatenate the embeddings of each feasible operation–machine pair, feed them into an MLP to obtain a score.

**Reward** Inspired by Zhang et al. (2020), we use an estimated lower-bound makespan as the reward[2]. Before the scheduling process begins, the lower-bound finish time for each operation can be computed iteratively as: $\underline{FT}_{ij} = \underline{FT}_{i(j-1)} + \min_{m \in \mathcal{M}_{ij}} D_{ij}^m$, and $\underline{FT}_{\max} = \max_{(ij) \in \mathcal{O}} \underline{FT}_{ij}$, where $\underline{FT}_{ij}$ denotes the lower-bound finish time of operation $O_{ij}$, and $\underline{FT}_{\max}$ represents the estimated lower-bound makespan. During the scheduling process, we compute the estimated lower-bound finish time $\underline{FT}_{\max}(s_t)$ based on the current state $s_t$. The reward at step $t$ is defined as the negative difference between the estimated lower-bound makespan before and after the action:

$$r_t = -(\underline{FT}_{\max}(s_{t+1}) - \underline{FT}_{\max}(s_t)). \tag{3}$$

### 4.2 POLICY NETWORK ARCHITECTURE

We design a decoder-only neural architecture (Drakulic et al., 2023) to solve scheduling problems, consisting of two main modules: a feature extraction network that models structural and temporal information into embeddings, and a lightweight MLP-based policy head that makes scheduling decisions. The overall architecture is illustrated in Figure 2.

#### 4.2.1 FEATURE EXTRACTION NETWORK

To effectively represent both structural and temporal aspects of the scheduling problem, we design a feature extraction network composed of two branches. The operation branch models O2O dependencies via self-attention, while the machine branch is built around cross-attention that aggregates operation information into machine embeddings. Both branches are built upon standard Transformer layers, with several targeted adaptations to better capture the unique scheduling characteristics.

**Operation branch with RoPE** In the absence of explicit positional encoding, intra-job order (i.e., O2O dependency) must be inferred implicitly across network layers, which is both inefficient and unreliable. To address this, we incorporate Rotary Positional Embedding (RoPE) (Su et al., 2024) into the *operation* branch to directly model relative intra-job distances without introducing additional learnable parameters, as shown in the middle bottom of Figure 2. Particularly, RoPE makes the similarity between query $q_a$ and key $k_b$ a function $g$ not only of their content embeddings $x_a$ and $x_b$, but also of their relative position $a - b$:

$$\langle \text{RoPE}_q(x_a, a), \text{RoPE}_k(x_b, b) \rangle = g(x_a, x_b, a - b). \tag{4}$$

---

[2]In Appendix F, we further compare this estimated lower-bound reward with two alternatives (estimated-mean and $\Delta$-makespan rewards).

In scheduling, dependencies arise only within individual jobs. Operations from different jobs or machines can be permuted arbitrarily without affecting decision-making, rendering their relative positions irrelevant. In this sense, RoPE is exclusively applied within the operation branch, injecting positional awareness into intra-job attention patterns. Unlike index-based positional features, RoPE has been shown to provide stronger generalization and better structural encoding. This allows us to simplify the feature set while still preserving the essential sequential information at the job level.

**Machine branch with Edge in Attention** The *machine* branch is designed to capture the structural interactions between operations and machines. A key aspect of this interaction is the processing duration, which is naturally defined on edges rather than belonging to either operation or machine nodes. To model this, we employ a cross-attention mechanism in which each operation attends to all of its candidate machines, incorporating both node-level and edge-level information. This design is illustrated in Figure 2 (upper branch of the Feature Extraction module).

Unlike prior cross-attention approaches (Kwon et al., 2021; Drakulic et al., 2024), which incorporate edge features indirectly by adjusting attention scores, our design integrates edge information directly by embedding it into the value vectors. This ensures that edge attributes influence not only the attention weights but also the final aggregated representations. Formally, for an operation node $O_{ij}$ and a machine node $M_m$, the attention is computed as:

$$\text{Attention}(M_m, O_{ij}) = \sigma \left( \frac{(q_m + q_{m,ij})^\top (k_{ij} + k_{m,ij})}{\sqrt{d}} \right) \cdot (v_{ij} + v_{m,ij}), \tag{5}$$

where $\sigma$ denotes the softmax function, $q_m$ is the query from the machine node, $k_{ij}$ and $v_{ij}$ are the key and value from the operation node, and $q_{m,ij}, k_{m,ij}, v_{m,ij}$ are edge-specific projections derived from the duration $D_{ij}^m$. Since the number of operation–machine pairs scales with $|\mathcal{O}| \times |\mathcal{M}|$, learning independent projection parameters for each edge would be computationally prohibitive. To maintain efficiency, we share projection weights across all attention heads as well as across the query, key, and value projections, significantly reducing parameter count and memory usage.

**Machine branch with Self-based Cross-attention** In scheduling problems, the number of operations often exceeds the number of machines by an order of magnitude. This structural asymmetry leads to a severe information imbalance: each machine must aggregate information from a disproportionately large number of operations, i.e., often 10 times or more, which dilutes attention signals and destabilizes training. Inspired by the machine-node embedding aggregation in Song et al. (2023), we introduce a self-based cross-attention mechanism, where each machine node also attends to its own representation during attention weight computation (i.e., upper branch of the Feature Extraction module in Figure 2). While residual connections inject self-information unconditionally after attention, the self-based formulation enables the model to assign a soft, adaptive attention weight to the machine's own embedding. This helps preserve critical machine-level information in the presence of overwhelming inter-node messages. Formally, for a machine node $M_m$ with projected value vector $v_m$ and operation projected value vectors $v_{ij} \in \mathbb{R}^d$, the attention output $h'_m$ is defined as:

$$h'_m = \alpha_{mm} v_m + \sum_{(ij) \in \mathcal{N}(M_m)} \alpha_{ij} v_{ij}, \tag{6}$$

where $\alpha_{mm}$ is the attention weight assigned to the machine node itself, $\alpha_{ij}$ are the attention weights for operation nodes connected to $M_m$, and $\mathcal{N}(M_m)$ denotes the set of such operation nodes. To reduce parameters, we share the query, key, and value projection weights across machine nodes.

### 4.2.2 DECISION-MAKING

Our decision-making module follows a standard policy network design adopted in prior DRL-based scheduling works (Song et al., 2023; Wang et al., 2024b; Zhao et al., 2025) based on operation-machine pairs. It consists of a multi-layer perceptron (MLP) that takes as input the operation and machine embeddings from the feature extraction network, along with the edge (i.e., duration) embeddings, and produces a scalar score for each feasible operation–machine pair. A softmax over these scores yields the final probability distribution. Notably, for simplicity, we do not include a global embedding in decision-making, as it shows limited effectiveness[3]. Additionally, as discussed

---

[3]We demonstrate in Appendix B.3, using DANIEL (Wang et al., 2024b) as an example, that removing its global embedding does not degrade performance.

in Section 4.1.2, we do not incorporate heuristic masking to prune the action space beyond the hard scheduling constraints (i.e., O2O dependency/O2M connection).

However, unlike most prior works (Zhang et al., 2020; Song et al., 2023; Wang et al., 2024b; Zhao et al., 2025) that adopt actor-critic frameworks such as Proximal Policy Optimization (PPO) (Schulman et al., 2017), we leverage a simple REINFORCE algorithm (Williams, 1992) to optimize the policy. Although vanilla REINFORCE is known to exhibit higher variance than actor–critic methods such as PPO, we deliberately adopt REINFORCE to isolate the impact of state and architecture design from algorithmic improvements. This choice simplifies implementation and aligns with standard Transformer-based NCO practice, such as AM (Kool et al., 2019) and POMO (Kwon et al., 2020). It keeps the training pipeline minimal, without additional critic networks or auxiliary losses, and allows us to focus on the impact of the proposed state representation and architecture rather than on the choice of specific RL algorithm. (However, we also implemented the PPO version of our method for a more comprehensive evaluation, which is presented in Appendix E.3). The reward is defined (in Eq. (3)) as the negative difference between the estimated lower-bound makespan before and after taking an action. Details of the training algorithm are provided in Appendix B.3.

## 5 EXPERIMENTS

In this section, we conduct extensive experiments on FJSP to demonstrate the effectiveness of RESCHED, comparing it with strong baselines and performing ablations on its key components. As a generic framework, we also extend our evaluation to JSSP and FFSP to showcase its generality.

**Training and Evaluation Settings** We train RESCHED on FJSP and two variants, JSSP and FFSP, respectively. For each problem, we generate one or two million instances for training, depends on the problem size. The models are trained on smaller problem sizes and evaluated on significantly larger ones as well as standard benchmarks: Bandimarte (Brandimarte, 1993) and Hurink (Hurink et al., 1994) for FJSP, Taillard (Taillard, 1993) and DMU (Demirkol et al., 1998) for JSSP, and extended sizes up to 100×12 for FFSP[4]. Notably, for both JSSP and FFSP, we use only a single training size (10×10 for JSSP and 20×12 for FFSP) to demonstrate generalization capability. Additionally, we evaluate the policies not only using a greedy strategy, but also with a sampling strategy. Following HGNN (Song et al., 2023) and DANIEL (Wang et al., 2024b), for each test instance we run 100 independent stochastic decoding trajectories of the policy in parallel, where at every decision step the next operation–machine pair is sampled from the network's categorical output, and we report the solution with the smallest makespan among the 100 trajectories. Further details of the dataset and configurations are provided in Appendix B.2. Our code is available at `https://github.com/XiangjieXiao/ReSched`.

**Baselines** We compare RESCHED against three groups of baselines. (i) Classical priority dispatching rules (PDRs), including FIFO, SPT, MOPNR, and MWKR. (ii) State-of-the-art DRL-based methods: HGNN (Song et al., 2023), DANIEL (Wang et al., 2024b), and DOAGNN (Zhao et al., 2025) for FJSP; L2D (Zhang et al., 2020) and RL-GNN (Park et al., 2021b) for JSSP; and Mat-Net (Kwon et al., 2021) for FFSP. (iii) Strong non-learning baselines, including the 2SGA genetic algorithm (Rooyani & Defersha, 2019) tailored for FJSP and the CP-SAT solver from OR-Tools (Da Col & Teppan, 2019), which we apply to both FJSP and JSSP benchmarks. More details are given in Appendix B.3.

### 5.1 PERFORMANCE ON FJSP

**In-Distribution Performance** Table 1 shows RESCHED outperforms all baselines on both SD$_1$ (Song et al., 2023) and SD$_2$ (Wang et al., 2024b) datasets, achieving superior results in **14/16** cases. The advantage is most pronounced on challenging SD$_2$ instances, where RESCHED reduces the gap by **30%** versus DANIEL (15×10 case). Even on simpler SD$_1$ instances, it maintains consistent improvements, cutting DANIEL's gap by half in 15×10 and 20×10 settings, demonstrating robust performance across difficulty levels.

---

[4]The notation $n \times m$ indicates $n$ jobs and $m$ machines(the number of operations per job varies across datasets and is omitted here for brevity).

Table 1: Results on datasets: in-distribution (top); out-of-distribution (middle); benchmark (bottom)

| Dataset | Size | | PDRs | | | | Greedy | | | Sampling | | | OR-Tools[1] |
|---|---|---|---|---|---|---|---|---|---|---|---|---|---|
| | | | FIFO | SPT | MOPNR | MWKR | HGNN | DANIEL | RESCHED | HGNN | DANIEL | RESCHED | |
| SD1 | 10×5 | Gap(%)↓ | 24.06 | 34.76 | 19.87 | 17.58 | 16.03 | **10.87** | 12.25 | 9.66 | **5.57** | 5.98 | 96.32 (5%) |
| | 20×5 | Gap(%)↓ | 14.87 | 22.56 | 13.85 | 11.51 | 12.27 | 5.03 | **4.63** | 10.31 | 2.46 | **2.33** | 188.15 (0%) |
| | 15×10 | Gap(%)↓ | 28.65 | 38.22 | 20.68 | 19.41 | 16.33 | 12.42 | **6.51** | 12.13 | 6.79 | **3.09** | 143.53 (7%) |
| | 20×10 | Gap(%)↓ | 19.22 | 30.25 | 12.20 | 10.30 | 10.15 | 1.31 | **0.48** | 9.64 | -1.03 | **-1.55** | 195.98 (0%) |
| SD2 | 10×5 | Gap(%)↓ | 76.47 | 57.96 | 72.52 | 70.01 | 71.42 | 25.68 | **16.36** | 49.71 | 12.57 | **6.39** | 326.24 (96%) |
| | 20×5 | Gap(%)↓ | 74.59 | 38.91 | 74.58 | 71.31 | 76.79 | 11.52 | **9.87** | 60.70 | 4.66 | **3.68** | 602.04 (0%) |
| | 15×10 | Gap(%)↓ | 132.23 | 86.74 | 125.32 | 121.45 | 115.26 | 57.16 | **18.14** | 101.52 | 38.70 | **9.81** | 377.17 (28%) |
| | 20×10 | Gap(%)↓ | 135.27 | 78.82 | 129.09 | 124.98 | 126.12 | 31.58 | **14.18** | 114.15 | 19.13 | **7.90** | 464.16 (1%) |

| Dataset | Size | | Top PDRs | | Greedy | | | | | | Sampling | | | | | | OR-Tools |
|---|---|---|---|---|---|---|---|---|---|---|---|---|---|---|---|---|---|
| | | | SPT | MWKR | HGNN | | DANIEL | | RESCHED | | HGNN | | DANIEL | | RESCHED | | |
| | | | | | 10×5 | 20×10 | 10×5 | 20×10 | 10×5 | 20×10 | 10×5 | 20×10 | 10×5 | 20×10 | 10×5 | 20×10 | |
| SD1 | 30×10 | Gap(%)↓ | 27.47 | 13.96 | 14.61 | 14.01 | 5.10 | **2.50** | 3.44 | 2.69 | 12.36 | 13.49 | 4.43 | 1.67 | 3.49 | **1.51** | 274.67 (6%) |
| | 40×10 | Gap(%)↓ | 21.66 | 13.37 | 14.21 | 13.75 | 3.65 | **1.52** | 2.54 | 1.64 | 12.26 | 13.49 | 3.77 | 1.14 | 3.64 | **1.10** | 365.96 (3%) |
| SD2 | 30×10 | Gap(%)↓ | 59.74 | 122.89 | 126.55 | 123.57 | 14.85 | 11.95 | 8.79 | **6.30** | 115.21 | 111.51 | 9.47 | 4.80 | 3.59 | **1.40** | 692.26 (0%) |
| | 40×10 | Gap(%)↓ | 38.74 | 108.66 | 109.87 | 108.12 | 0.52 | -1.67 | -2.40 | **-4.58** | 102.45 | 99.26 | -2.74 | -6.60 | -5.69 | **-7.61** | 998.39 (0%) |

| Strategy | Dataset | | MWKR (Top PDR) | HGNN 10×5 | HGNN 15×10 | DANIEL 10×5 | DANIEL 15×10 | DOAGNN 10×5 | RESCHED 10×5 | RESCHED 15×10 | 2SGA | OR-Tools | UB[2] |
|---|---|---|---|---|---|---|---|---|---|---|---|---|---|
| Greedy | Brandimarte | Gap(%)↓ | 28.91 | 28.52 | 26.77 | 13.58 | 12.97 | 31.64 | **9.08**[3] | 12.49 | 175.20(3.17%) | 174.20(1.5%) | 172.7 |
| | Hurink(edata) | Gap(%)↓ | 18.6 | 15.53 | 15.0 | 16.33 | **14.41** | 16.21 | 15.48 | 16.34 | - | 1028.93(-0.03%) | 1028.88 |
| | Hurink(rdata) | Gap(%)↓ | 13.86 | 11.15 | 11.14 | 11.42 | 12.07 | 11.83 | **10.18** | 10.31 | - | 935.80(0.11%) | 934.28 |
| | Hurink(vdata) | Gap(%)↓ | 4.22 | 4.25 | 4.02 | 3.28 | 3.75 | 4.32 | 3.48 | **2.55** | 812.20(0.39%) | 919.60(-0.01%) | 919.50 |
| Sampling | Brandimarte | Gap(%)↓ | 28.91 | 18.56 | 19.0 | 9.53 | 8.95 | 18.62 | **6.61** | 8.14 | 175.20(3.17%) | 174.20(1.5%) | 172.7 |
| | Hurink(edata) | Gap(%)↓ | 18.6 | 8.17 | 8.69 | 9.08 | 8.72 | 8.46 | **8.13** | 10.39 | - | 1028.93(-0.03%) | 1028.88 |
| | Hurink(rdata) | Gap(%)↓ | 13.86 | 5.57 | 5.95 | 4.95 | 5.49 | 5.83 | 5.04 | **4.92** | - | 935.80(0.11%) | 934.28 |
| | Hurink(vdata) | Gap(%)↓ | 4.22 | 1.32 | 1.34 | **0.69** | 0.72 | 1.44 | 0.82 | **0.69** | 812.20(0.39%) | 919.60(-0.01%) | 919.50 |

1. OR-Tools (1800s per instance): solution and optimal ratio reported;
2. UB is the best-known solution (Behnke & Geiger, 2012), used as the baseline to compute gaps;
3. **Instance-wise average gap** is reported to reduce bias from varying instance scales.

Table 3: Results on Taillard Benchmark for JSSP.

| Size | PDRs | | | | DRL-based | | | OR-Tools | UB |
|---|---|---|---|---|---|---|---|---|---|
| | SPT | MWKR | FDD/MWKR | MOPNR | L2D | RL-GNN | ReSched 10×10 | | |
| 15×15 | 54.8 | 56.7 | 47.1 | 45.0 | 26.0 | 20.1 | **15.74** | 0.1 | 1233.9 |
| 20×15 | 65.2 | 60.7 | 50.6 | 47.7 | 30.0 | 24.9 | **19.7** | 0.2 | 1361.3 |
| 20×20 | 64.2 | 55.7 | 47.6 | 42.8 | 31.6 | 29.2 | **16.3** | 0.7 | 1617.1 |
| 30×15 | 61.6 | 52.6 | 45.0 | 45.6 | 33.0 | 24.7 | **21.5** | 2.1 | 1771.2 |
| 30×20 | 66.0 | 63.9 | 56.3 | 48.2 | 33.6 | 32.0 | **22.5** | 2.8 | 1919.4 |
| 50×15 | 51.4 | 40.9 | 34.8 | 30.1 | 22.4 | 15.9 | **16.1** | 0.0 | 2783.8 |
| 50×20 | 59.5 | 53.9 | 41.5 | 37.9 | 26.5 | 21.3 | **15.6** | 2.8 | 2834.4 |
| 100×20 | 41.0 | 32.9 | 23.4 | 20.2 | 13.6 | **9.2** | 9.6 | 3.9 | 5369.6 |

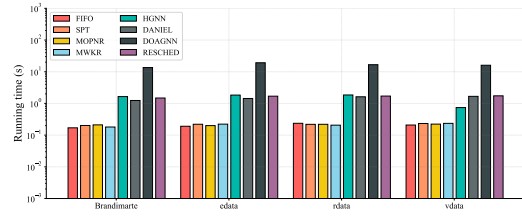

Figure 3: Running Time on FJSP Benchmark

**Generalization Performance** RESCHED demonstrates strong generalization on both synthetic and benchmark datasets (Table 1). On larger synthetic instances (30×10 to 40×10), it outperforms DRL baselines in **6/8** cases, even surpassing OR-Tools by **7.61%** on the challenging SD2 with 40×10 setting. For Brandimarte and Hurink benchmarks, trained solely on SD1, RESCHED achieves best performance in **7/8** cases across both strategies. Notably, the DOAGNN does not report results for the 15×10 setting, and is thus only compared using one configuration. Compared with strong non-learning baselines, 2SGA and OR-Tools (with 3600s time limit) can often obtain solutions very close to the best-known upper bounds, but at the cost of substantially longer computation time due to lengthy population-based search or exact optimization. In contrast, once trained, RESCHED produces competitive solutions within a short inference time per instance (see Figure 3), and its performance can be further improved by simple sampling-based decoding without additional training cost, making it more suitable for time-sensitive or repeatedly solved scheduling scenarios. These results suggest RESCHED's potential for generalization to real-world settings with complex characteristics under practical time budgets.

**Running Time Analysis** To evaluate runtime efficiency, we conduct experiments on open benchmark instances characterized by diverse problem structures. Each algorithm is independently executed five times to ensure reliable results, and their average running times are reported in Figure 3. Notably, our method achieves a runtime comparable to existing DRL-based approaches while outperforming current state-of-the-art methods in terms of scheduling quality.

## 5.2 ABLATION STUDY AND ROBUSTNESS ANALYSIS

We conduct ablation studies on RESCHED's two key innovations: (1) the minimal representation and (2) attention-based architectural improvements, by removing each design element individually.

Results in Table 5 (Appendix B.3) confirm that all components contribute positively to performance. Additionally, we perform robustness analysis by training RESCHED across multiple random seeds to assess its stability and consistency. The results, presented in Table 6 (Appendix B.3), demonstrate that the variation across seeds is minimal (less than 1%), further confirming that RESCHED's improvements are both robust and consistent.

## 5.3 PERFORMANCE ON JSSP AND FFSP

We evaluate RESCHED's generalization capability on the Taillard benchmark (Taillard, 1993) for JSSP. Trained solely on synthetic $10 \times 10$ instances (generated under the same distribution as Zhang et al. (2020)), our model is directly tested on benchmark instances ranging from $15 \times 15$ to $100 \times 20$ by greedy strategy. The results in Table 3 show that, RESCHED outperforms L2D[5] and RL-GNN in 7 out of 8 test sizes, even surpassing their in-distribution performance (trained and tested on the same size). For reference, we also report the results of the CP-SAT solver in OR-Tools (Da Col & Teppan, 2019) with a 3600-second time limit per instance, which provides strong upper bounds on the Taillard instances. This demonstrates exceptional scalability, as no size-specific tuning is required, which exhibits our framework's ability to adapt to different scheduling problems. Results on in-distribution settings and the DMU benchmark, when evaluated with a greedy decoding strategy, also confirmed its consistent superiority (see Appendix B.3). We further evaluate RESCHED on FFSP under MatNet's setting (Kwon et al., 2021). Unlike MatNet, which trains a separate model for each size (20/50/100), RESCHED, trained only on size 20, achieves the best results in most settings across sizes under both greedy and sampling strategies (Table 9). Same as MatNet, we also use 24 parallel solutions per instance, which showed superior generalization.

## 6 CONCLUSION

In this paper, we present RESCHED, a novel framework for solving scheduling problems using deep reinforcement learning. RESCHED introduces a simplified state representation and a Transformer-based architecture, which effectively captures the structural and temporal characteristics of scheduling problems. Our extensive experiments on FJSP, JSSP, and FFSP demonstrate that RESCHED achieves favorable performance while maintaining high efficiency. The results highlight the potential of RESCHED as a generic framework for various scheduling tasks. However, RESCHED currently encodes only O2O/O2M interaction with self-attention/unidirectional cross-attention; explicit machine to operation (M2O) feedback is not yet modeled and will be explored in future. Moreover, we will also investigate how to adapt our method to search-based works (Zhang et al., 2024).

## ACKNOWLEDGEMENTS

This research is supported by the National Research Foundation, Singapore under its AI Singapore Programme (AISG Award No: AISG3-RP-2022-031). This work is also supported by the National Natural Science Foundation of China (No. 62473233).

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

# A STATE REPRESENTATION: FROM FJSP TO OTHER VARIANTS

## A.1 REVISITING THE FSJP FORMULATION

In FJSP, for each operation node $O_{ij}$, we define:

- $ST_{ij}$: the start time of operation $O_{ij}$;
- $D_{ij}^m$: the duration of operation $O_{ij}$ on machine $m$;
- $FT_{ij}$: the finish time of operation $O_{ij}$.

For each machine node $M_m$, we define:

- $AT_t^m$: the available time of machine $m$ at the current scheduling step.

Whenever $a_{t,ij}^m = 1$, operation node $O_{ij}$ may start only satisfies the following two constraints: the operation dependency constraints and the machine availability constraints. Then the finish time $FT_{ij}$ and start time $ST_{ij}$ can be computed as follows:

$$FT_{ij} = ST_{ij} + D_{ij}^m, \quad \text{if } a_{t,ij}^m = 1$$
$$ST_{ij} = \max\left(FT_{i(j-1)}, AT_t^m\right) \tag{7}$$

where $FT_{i(j-1)}$ is the finish time of predeccessor operation $O_{i(j-1)}$ in the job $J_i$.

For the machine nodes, the available time $AT_t^m$ at step $t$ is updated as follows:

$$AT_t^m = \begin{cases} FT_{i'j'} & \text{if } a_{t-1,i'j'}^m = 1 \\ AT_{t-1}^m & \text{otherwise} \end{cases} \tag{8}$$

where $(i'j')$ is the operation assigned to machine $m$ at step $t-1$.

Finally, for a scheduling problem like FJSP, we aim to optimize the solution $\mathcal{A}$ to minimize the makespan $FT_{\max}$, which is defined as the maximum finish time across all operations:

$$FT_{\max} = \max_{(ij) \in \mathcal{O}} FT_{ij}. \tag{9}$$

**Remark 1.** *As we analysed in the Proposition 1, it is unnecessary to explicitly retain the full history of past states. This means we do not need to directly track the finish times of operations scheduled in previous steps. However, since the finish time of an operation $O_{ij-1}$ serves as the available time for its successor $O_{ij}$, we can instead maintain the **operation available time** a quantity that captures the same information in a recursive manner. Thus the Eq. (7) can be simplified as Eq. (2) in the main text.*

Using the above formulation, for a given FJSP instance and feasible solution $\mathcal{A}$, we can compute each operation's status and machine's status at each scheduling step.

## A.2 FROM FJSP TO OTHER SCHEDULING PROBLEMS

From heterogeneous graph perspective, FJSP provides a unified formulation that naturally extends to two classical variants: JSSP and FFSP.

**JSSP as a special case.** In JSSP, each operation is tied to exactly one machine rather than a set of machines. Consequently, the duration $D_{ij}^m$ and schedule $a_{t,ij}^m$ degenerate to $D_{ij}$ and $a_{t,ij}$, respectively. The O2O dependencies remain unchanged, whereas the O2M connections become one-hot. The state representation and update rules are the same as in FJSP, with trivial O2M connections.

**FFSP as a special case.** In FFSP, all jobs follow an identical sequence of stages, i.e. an identical routing with one operation per stage. In this context, an "operation" can be viewed as a *stage*. Each stage $j$ is executed at a *station*, and under the flexible setting, a station is typically composed of multiple parallel machines capable of performing the same task. Hence, FFSP can be viewed as an FJSP with $\mathcal{M}_{ij} = \mathcal{M}_j$ for all jobs $i$. The O2O dependencies reduce to stage-to-stage precedence, while the O2M connections remain similar to FJSP, where each stage is connected to its corresponding station's machines.

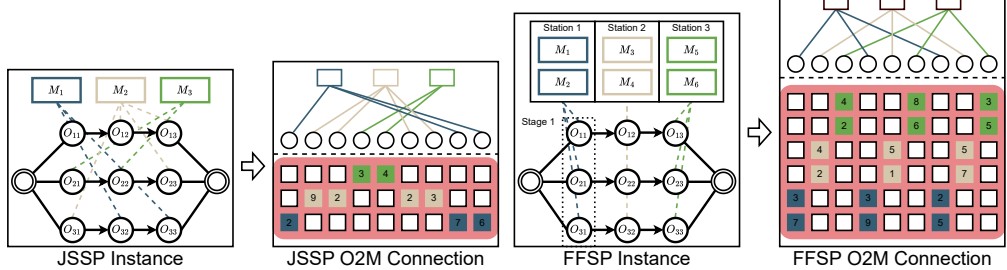

Figure 4: The illustration of JSSP and FFSP instances, respectively.

**Implications for our framework.**   Across these variants, the core state representation and update rules remain consistent with those of FJSP. The O2O dependencies are preserved; differences arise only in the pattern of O2M connections, reflecting each problem's machine-availability constraints. Figure 4 illustrates both variants with identical O2O dependencies and differing machine-availability constraints.

# B    TRAINING ALGORITHM AND EXPERIMENT

## B.1    TRAINING ALGORITHM

We introduce the training algorithm for RESCHED in Algorithm 1. The training process follows the REINFORCE algorithm, where we sample actions from the policy network and compute the policy loss based on the rewards received. The model parameters are updated using gradient descent.

---

**Algorithm 1** Training RESCHED with REINFORCE

---

1: **Input:** Scheduling environment $\mathcal{E}$, model parameters $\theta$, number of epochs $E$, training episodes $N$, batch size $B$, learning rate $\alpha$;
2: **for** epoch $e = 1$ to $E$ **do**
3:     Initialize score and loss meter;
4:     **while** $ep < N$ **do**
5:         Generate $B$ instances using environment $\mathcal{E}$;
6:         Reset the environment to get initial state $s_0$;
7:         Initialize empty trajectory: $\mathcal{T} = \emptyset$;
8:         **repeat**
9:             Process current state $s_t$ into model input;
10:             Sample action $a_t \sim \pi_\theta(\cdot \mid s_t)$;
11:             Execute $a_t$ to get reward $r_t$ and next state $s_{t+1}$;
12:             Store $(s_t, a_t, r_t, s_{t+1})$ into $\mathcal{T}$;
13:             $s_t \leftarrow s_{t+1}$;
14:         **until** task is finished
15:         Compute return $G_t$ using discounted cumulative rewards;
16:         Normalize advantages: $A_t = G_t - \text{mean}(G_t)$;
17:         Compute policy loss $\mathcal{L} = -\sum_t A_t \log \pi_\theta(a_t \mid s_t)$;
18:         Update parameters: $\theta \leftarrow \theta - \alpha \nabla_\theta \mathcal{L}$;
19:         $ep \leftarrow ep + B$;
20:     **end while**
21:     Validate $\pi_\theta$ on validation set;
22: **end for**
23: **Output:** Trained model parameters $\theta$

---

## B.2 EXPERIMENT DETAILS

**Datasets**  For FJSP, the model is trained on synthetic datasets $SD_1$ (Song et al., 2023) and $SD_2$ (Wang et al., 2024b), to evaluate its in-distribution performance, and then evaluated on larger size as well as standard benchmark: Bandimarte (Brandimarte, 1993) and Hurink (Hurink et al., 1994). The synthetic datasets are generated using the same method as in Song et al. (2023) and Wang et al. (2024b). Specifically, for an instance of FJSP with $n$ jobs and $m$ machines:

- $SD_1$: Duration $D_{ij}^m$ is uniformly sampled from $[1, 20]$; Each job's operations number is uniformly sampled from $[0.8n, 1.2n]$;
- $SD_2$: Duration $D_{ij}^m$ is uniformly sampled from $[1, 99]$; Each job's operations number is uniformly sampled from $[1, n]$.

For JSSP, the model is trained on the synthetic dataset to evaluate its in-distribution performance, and then evaluated on the standard benchmark: Taillard (Taillard, 1993) and DMU (Demirkol et al., 1998). The synthetic dataset is generated as follows: Duration $D_{ij}$ is uniformly sampled from $[1, 99]$; Each job's operations number is uniformly sampled from $[1, n]$.

For FFSP, the model is trained on the synthetic dataset generated with the same method as in Kwon et al. (2021), to evaluate its in-distribution performance and cross-size generalization performance. The synthetic dataset is generated as follows: Duration $D_{ij}^m$ is uniformly sampled from $[2, 9]$; Each job has 3 stages, and each stage has 4 parallel machines as a station.

**Configurations**  The RESCHED framework is implemented in PyTorch (Paszke et al., 2019) and trained using the REINFORCE algorithm (Williams, 1992). The feature extraction network consists of 2-layer Transformer blocks, each with 8 attention heads, a hidden dimension of 128, and a feed-forward dimension of 512. The decision-making network is a 3-layer MLP, with each layer containing 64 hidden units, following prior works (Zhang et al., 2020; Song et al., 2023; Wang et al., 2024b; Zhao et al., 2025). We use the Adam optimizer with a learning rate of $5 \times 10^{-5}$. The model is trained for 2000 epochs, with 1000 training instances per epoch and a batch size of 50. The discount factor $\gamma$ is set to 0.99. Due to resource constraints, for lager sized problem, the number of training instances per epoch is reduced to 500 and the batch size to 24. During training, we use the estimated lower bound of the makespan as the reward, as described in Eq. (3), with a discount factor of 0.99. The model that achieves the best performance on the validation set is saved and later evaluated on the test set (100 generated instances) and benchmark datasets. We use a single NVIDIA RTX A40 GPU for training and evaluation. We will release the code and data generation scripts.

**Hyperparameter tuning and reporting**  We tune RESCHED on the $SD_1$ $10 \times 5$ setting and *keep the selected hyperparameters fixed* for all other datasets and sizes. For baselines, we do not perform additional tuning beyond the authors' default settings. We either directly report the results from their original papers or use their open-source code with the provided hyperparameters and checkpoints to ensure a fair comparison. All baselines have open-source implementations; some also provide checkpoints.

**Baselines**  To assess the effectiveness of RESCHED, we compare it against both rule-based and DRL-based baselines commonly adopted in the FJSP literature. The baselines are grouped into two categories:

**(1) Priority Dispatching Rules (PDRs).** We include four widely used heuristic rules:

- **FIFO** (First-In-First-Out): Selects the earliest operation–machine pair based on the order of arrival.
- **SPT** (Shortest Processing Time): Selects the pair with the shortest operation duration.
- **MOPNR** (Most Operations Remaining): Selects the pair associated with the job that has the largest number of remaining operations, breaking ties by the earliest machine available time.
- **MWKR** (Most Work Remaining): Selects the pair associated with the job that has the largest total remaining processing time, using the average duration of successor operations, and breaks ties by the earliest machine available time.

These heuristics are widely adopted due to their efficiency, simplicity, and strong generalization ability, particularly in large-scale or unseen scheduling instances (Sels et al., 2012). We implement these PDRs using the open-source code by Song et al. (2023) and keep the same hyperparameter settings.

**(2) Neural methods for FJSP** We also compare RESCHED against three representative GNN-based approaches developed for FJSP:

- **HGNN** (Song et al., 2023): Models the scheduling problem as a heterogeneous graph, where operations and machines are treated as distinct node types.
- **DANIEL** (Wang et al., 2024b): Employs a dual-attention mechanism to jointly capture operation–machine interactions.
- **DOAGNN** (Zhao et al., 2025): Leverages a decoupled disjunctive-graph formulation to better encode precedence and machine constraints.

All three methods are DRL frameworks based on GNN or variants (e.g., GAT) (Velickovic et al., 2017), and using the PPO (Schulman et al., 2017) as training algorithm. For HGNN and DANIEL, we directly report the results from their original papers. For DOAGNN, we use their open-source code and evaluate it on the open benchmark using their provided checkpoints. Additionally, we retrain DANIEL from scratch in our ablation study to validate our simplified feature set, maintaining the same hyperparameter settings as in the original work.

**(3) Neural methods for other variants** Additionally, we compare RESCHED against two GNN-based methods targeting JSSP; and a transformer-based method for FFSP:

- **L2D** (Zhang et al., 2020): Proposes an end-to-end DRL framework that learns size-agnostic priority dispatching rules for JSSP, based on disjunctive graph representation and Graph Isomorphism Networks.
- **RL-GNN** (Park et al., 2021b): Integrates RL with a GNN-based encoder-decoder network to solve JSSP, capturing both job and machine contexts through dynamic graph representations.
- **Matnet** (Kwon et al., 2021): Introduces a matrix-based encoding of combinatorial structures for routing problems and FFSP, enabling flexible attention across decision steps. It applies a transformer-style architecture to scheduling by encoding instance states as 2D matrices and training via REINFORCE.

For L2D and RL-GNN, we directly report the results from their original papers. For Matnet, we use their open-source code to retrain and evaluate it under the default hyperparameters.

## B.3 EXPERIMENTAL RESULTS

**Ablation Study on DANIEL** The input embedding for decision-making module in DANIEL is the concatenation of the *operation*, *machine*, *pair* and *global* embeddings, involving a total of **26** features (10 for operation, 8 for machine, 8 for pair); the global embedding is learned and does not add additional raw features. To evaluate the effectiveness of each component, we conduct an ablation study by removing the components one by one. The experiments are conducted on the synthetic dataset $SD_2$ with $15 \times 10$, and the results are shown in Table 4.

Table 4: Ablation study on DANIEL.

| setting | Obj.↓ | Avg. $\Delta(\%)$ ↓ |
|---|---|---|
| DANIEL | 589.44 | 56.28 |
| Del Global | 589.12 | 56.19 |
| Del Global MA | 588.66 | 56.07 |
| Del Gloabl MA P(-P6) | 588.76 | 56.10 |
| Del Gloabl MA P(-P6) O(-O2379) | 587.68 | 55.81 |

We conduct an ablation study on DANIEL to analyze the impact of different input embeddings. The results in Table 4 reveal the following insights:

- **Del Global:** Removing the *global* embedding leads to a slight improvement, indicating that this component may introduce redundancy or noise.
- **Del Global MA:** Further removing the *machine* embedding, comprising 8 machine-related features, results in continued improvement, suggesting that the model can perform well without explicit machine descriptors.
- **Del Global MA P(-P6):** Based on the previous experiment, we further prune the *pair-wise* features from 8 to a single (6th) feature, which still maintains performance. This indicates that most pair features are not essential for effective scheduling.
- **Del Global MA P(-P6) O(-O2379):** Afterwards, based on all the previous experiments, we prune the *operation* features from 10 to 4 (retaining only features 2, 3, 7, and 9), and the performance is still comparable to the original DANIEL. This further validates that the full feature set is not strictly necessary for effective scheduling.

In the last step above, we eliminate the number of features in DANIEL *from 26 to only 5* (4 for operation, i.e. O2379, and 1 for pair-wise, i.e. P6), and the performance is still comparable to the original DANIEL.

Table 5: Ablation study on *ReSched*.

| Ablation | | In-Distribution $10 \times 5$ | Out-of-Distribution $30 \times 10$ | $40 \times 10$ | Open Benchmark Brandimarte | Avg. $\Delta(\%) \downarrow$ [1] |
|---|---|---|---|---|---|---|
| **RESCHED** | Gap(%)↓ | **12.25** | 3.44 | **2.54** | **9.08** | **+0.00** |
| | $\Delta(\%) \downarrow$ | 0.00 | +0.10 | 0.00 | 0.00 | |
| **Connection** | Gap(%)↓ | 16.42 | 6.74 | 3.83 | 14.48 | +3.54 |
| | $\Delta(\%) \downarrow$ | +4.17 | +3.30 | +1.29 | +5.40 | |
| **Relative Available Time** | Gap(%)↓ | 13.33 | 5.02 | 3.77 | 14.84 | +2.41 |
| | $\Delta(\%) \downarrow$ | +1.08 | +1.58 | +1.23 | +5.76 | |
| **Current Time** | Gap(%)↓ | 13.49 | 3.38 | 2.73 | 12.71 | +1.25 |
| | $\Delta(\%) \downarrow$ | +1.24 | -0.06 | +0.19 | +3.63 | |
| **RoPE** | Gap(%)↓ | 12.81 | **3.34** | 2.75 | 14.97 | +1.64 |
| | $\Delta(\%) \downarrow$ | +0.56 | -0.10 | +0.21 | +5.89 | |
| **Edge in Att** | Gap(%)↓ | 12.56 | 4.28 | 3.44 | 13.02 | +1.50 |
| | $\Delta(\%) \downarrow$ | +0.31 | +0.84 | +0.90 | +3.94 | |
| **Self-based CA** | Gap(%)↓ | 13.16 | 4.67 | 3.72 | 16.10 | +2.59 |
| | $\Delta(\%) \downarrow$ | +0.91 | +1.23 | +1.18 | +7.02 | |

$\Delta(\%)$ indicates the gap deviation from standard RESCHED.
Avg. $\Delta$ summarizes overall performance deterioration across all datasets.

**Ablation Study on RESCHED**   To evaluate the effectiveness of each proposed component in RESCHED, we conduct comprehensive ablation studies to analyze our simplified state representation and network architecture. Specifically, we individually remove six key ideas from our standard framework, including simplifications and attention-related improvements. The ablation settings are illustrated as follows:

- **Connection:** Removing the O2O connection, and using the original bidirectional connection, which are widely used in previous works (Zhang et al., 2020; Song et al., 2023; Wang et al., 2024b; Zhao et al., 2025).
- **Relative Available Time:** Replacing the relative available time with the absolute available time (for operation and machine).
- **Current Time:** Using the absolute current time to prune the action space as in previous works (Song et al., 2023; Wang et al., 2024b; Zhao et al., 2025).
- **RoPE:** Removing the RoPE mechanism in operation branch.
- **Edge in Att:** Removing the edge features in cross-attention mechanism in machine branch.
- **Self-based CA:** Removing the self-based cross-attention mechanism in machine branch.

All variants are trained on the smallest $SD_1$ dataset (10×5) and evaluated on four test settings, including in-distribution, out-of-distribution (30×10 and 40×10), and a challenging open benchmark (Bandrimarte). As shown in Table 5, each component contributes to the overall performance, confirming the effectiveness and necessity of our design choices.

Table 6: Performance of RESCHED trained with REINFORCE across four random seeds.

| Dataset | DANIEL | RESCHED-Avg. | RESCHED-Orig. | Seed1 | Seed2 | Seed3 |
|---|---|---|---|---|---|---|
| **In-distribution** | | | | | | |
| $SD_1$-10×05 | **10.87** | 11.75 | 12.25 | 11.40 | 12.33 | 11.03 |
| **Out-of-distribution** | | | | | | |
| $SD_1$-30×10 | 5.10 | **3.10** | 3.44 | 2.46 | 3.39 | 3.09 |
| $SD_1$-40×10 | 3.65 | **2.23** | 2.54 | 1.64 | 2.46 | 2.26 |
| $SD_2$-30×10 | 14.85 | **9.04** | 8.79 | 9.35 | 9.78 | 8.22 |
| $SD_2$-40×10 | 0.52 | **-2.61** | -2.40 | -3.00 | -1.80 | -3.23 |
| Avg. | 6.03 | **2.94** | 3.09 | 2.61 | 3.46 | 2.59 |
| **Open Benchmark** | | | | | | |
| Brandimarte | 13.58 | **10.28** | 9.08 | 11.70 | 10.10 | 10.24 |
| Hurink (edata) | 16.33 | **15.84** | 15.48 | 16.46 | 15.52 | 15.88 |
| Hurink (rdata) | 11.42 | **10.13** | 10.18 | 10.30 | 10.46 | 9.57 |
| Hurink (vdata) | 3.28 | **2.90** | 3.48 | 2.60 | 2.55 | 2.95 |
| Avg. | 11.15 | **9.78** | 9.56 | 10.27 | 9.66 | 9.66 |

Values are average optimality gaps (%) w.r.t. the best-known upper bounds.
"RESCHED-Avg." denotes the mean over all four RESCHED runs (original + three additional seeds).

**Robustness Evaluation under Multiple Random Seeds**  To verify the robustness of our framework, we reran RESCHED (REINFORCE version) on the $SD_1$-10×5 training setting with three additional independent runs using different random seeds (four runs in total, including the originally reported in Table 1). For each run, we evaluated the policy across in-distribution, out-of-distribution, and open benchmark test settings to comprehensively test the performance of RESCHED. For ease of comparison, Table 6 reports in the second column the average over all four independent runs (the originally reported run plus the three new ones), while the subsequent columns list the results of each individual run. On out-of-distribution and open benchmark tests, all RESCHED runs outperform DANIEL, while in-distribution performance on $SD_1-10×5$ remains comparable. Moreover, the variation across seeds is much smaller than the performance gap between RESCHED and DANIEL, indicating that our gains are robust and not due to a fortunate choice of random seed.

**Performance on JSSP with Synthetic Data and DMU Benchmark**  In the main text, we have reported the performance of RESCHED on the Taillard benchmark. Here, we also evaluate its performance on synthetic JSSP data and the DMU benchmark. The synthetic data is generated using the same method as in Zhang et al. (2020), and the DMU benchmark is a widely used benchmark for JSSP. The results are shown in Table 7 and Table 8, respectively. Similar to the Taillard benchmark, RESCHED achieves the best performance on both synthetic data and DMU benchmark by solely using the **same** model trained on the $10 \times 10$ synthetic data.

Table 7: Performance on synthetic datasets of JSSP.

| Size | PDRs | | | | DRL-based | | Opt. Rate(%)[1] |
|---|---|---|---|---|---|---|---|
| | SPT | MWKR | FDD/MWKR | MOPNR | L2D | RESCHED $10 \times 10$ | |
| $6 \times 6$ | 42.0 | 34.6 | 24.0 | 29.2 | 17.7 | **7.0** | 100 |
| $10 \times 10$ | 50.0 | 42.6 | 36.6 | 36.5 | 22.3 | **9.5** | 100 |
| $15 \times 15$ | 59.2 | 52.6 | 45.1 | 42.6 | 26.7 | **14.3** | 99 |
| $20 \times 20$ | 62.0 | 58.6 | 49.6 | 45.5 | 29.0 | **15.0** | 4 |
| $30 \times 20$ | 65.3 | 58.7 | 48.6 | 44.7 | 29.2 | **16.0** | 12 |
| $50 \times 20$ | 54.9 | 48.1 | 38.4 | 33.7 | 22.1 | **12.8** | 48 |
| $100 \times 20$ | 35.1 | 27.0 | 19.6 | 14.7 | 9.4 | **4.3** | 2 |

Opt. Rate is the rate of instances for which OR-Tools returns the optimal solution.

Table 8: Performance on DMU benchmark of JSSP.

| Size | PDRs | | | | DRL-based | | UB[1] |
|------|------|------|------|------|------|------|------|
| | SPT | MWKR | FDD/MWKR | MOPNR | L2D | RESCHED $10 \times 10$ | |
| $20 \times 15$ | 64.1 | 62.1 | 53.6 | 49.2 | 39.0 | **23.7**[2] | 3023.8 |
| $20 \times 20$ | 64.6 | 58.2 | 52.5 | 45.2 | 37.7 | **22.2** | 3472.6 |
| $30 \times 15$ | 62.6 | 60.9 | 54.1 | 47.1 | 41.9 | **27.8** | 3879.0 |
| $30 \times 20$ | 65.9 | 63.2 | 60.1 | 52.0 | 39.5 | **28.3** | 4248.4 |
| $40 \times 15$ | 55.9 | 52.9 | 51.4 | 44.7 | 35.4 | **26.5** | 4871.2 |
| $40 \times 20$ | 63.0 | 61.1 | 55.5 | 49.2 | 39.4 | **29.2** | 5240.9 |
| $50 \times 15$ | 50.38 | 48.94 | 52.55 | 40.78 | 36.2 | **26.3** | 5950.6 |
| $50 \times 20$ | 62.2 | 56.4 | 57.3 | 49.6 | 38.8 | **31.8** | 6227.3 |

1.Upper Bound (UB) is the best known solution (available) for each instance.
2.**Instance-wise average gap** is reported to provide a higher accuracy.

Table 9: Performance on FFSP.

| setting | Greedy | | | Sampling | | |
|---------|--------|--------|---------|----------|--------|---------|
| | FFSP20 | FFSP50 | FFSP100 | FFSP20 | FFSP50 | FFSP100 |
| Matnet20 | 28.05 | 52.58 | 93.00 | 27.31 | 52.36 | 93.40 |
| Matnet50 | 27.78 | 52.05 | 92.17 | 27.05 | 51.55 | 91.86 |
| Matnet100 | 27.64 | 51.79 | **91.79** | 27.09 | 51.40 | 91.50 |
| ReSched20 | **26.65** | **51.24** | 92.28 | **25.12** | **49.65** | **90.80** |

**Performance on FFSP**    In the main text, we reported the overall performance of RESCHED on the FFSP. Here, we present the detailed results in Table 9.

## C   THEORETICAL ACCELERATION OPPORTUNITIES: KV CACHE

The RESCHED framework allows for theoretical acceleration via a Key-Value (KV) caching mechanism, owing to the following structural properties:

- *Backward-looking edges:* Each operation node is updated *only* by its successor operation;
- The representation of each successor operation is *fixed* (determined by the minimum duration);
- Operation nodes are *not* updated by machine nodes, which contain dynamic features that vary across scheduling steps.

As illustrated in Figure 5, this structure enables us to cache and reuse the key-value pairs of previously computed operation nodes during multi-step decoding, avoiding redundant computation across steps. To see this, consider an instance with $n$ operations and $m$ machines, decoded in $n$ decision steps by an $L$-layer Transformer. Without KV cache, at each step the O2O branch performs self-attention over $n$ operation nodes, with complexity $\mathcal{O}(Ln^2)$, and the O2M branch performs cross-attention between $m$ machine queries and $n$ operation keys/values, with complexity $\mathcal{O}(Lmn)$. Over $n$ decoding steps, the total attention cost is therefore

$$\mathcal{O}\big(L(n^3 + mn^2)\big).$$

With KV cache, the operation representations (and their keys/values) are computed only once in the O2O branch, with cost $\mathcal{O}(Ln^2)$, and then reused at all later steps; the O2M branch still needs to be recomputed at each step due to changing machine availability, giving a total cost of

$$\mathcal{O}\big(Ln^2(1 + m)\big).$$

The asymptotic reduction in attention computation is thus by a factor of

$$\frac{L(n^3 + mn^2)}{Ln^2(1 + m)} = \frac{n + m}{1 + m},$$

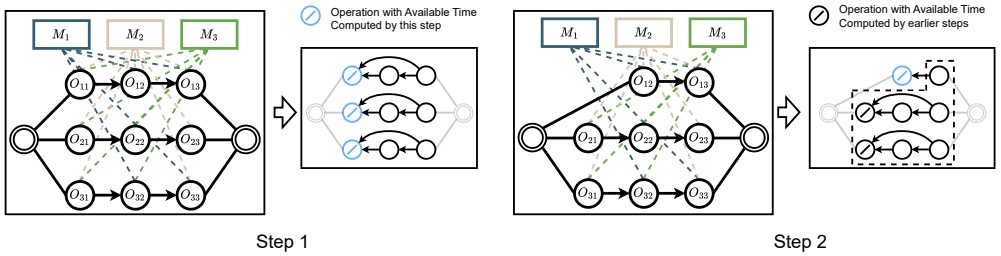

Figure 5: Illustration of the KV cache mechanism in RESCHED.

which approaches $\frac{n}{1+m}$ when $n \gg m$, a common regime in scheduling where the number of operations greatly exceeds the number of machines. In our experimental settings, the number of operations is typically 10–40 times larger than the number of machines (i.e., $n \approx 10m$–$40m$), which implies a reduction of about 10–40$\times$ in attention computation when KV cache is applied.

While this caching mechanism is not yet implemented in our current experiments, it presents a promising future direction for further inference-time acceleration.

## D  PROOF OF STATE-DEPENDENT OPTIMALITY IN SCHEDULING

*Proof.* Fix any decision step $t$ and any state $\mathcal{S}_t \in \mathbb{S}$. By Definition 4.1, $\mathcal{S}_t$ uniquely specifies the following elements: (i) the available time $AT_t^m$ of each machine $m$; (ii) for every job $j$, the completion status of its operations up to $t$, which is equivalent to knowing the finish time of the most recently scheduled operation of $j$; (iii) the operation–operation dependency (O2O); and (iv) the operation–machine feasibility graph with duration (O2M).

Let $\mathcal{O}$ be the set of all operations in the scheduling instance described by $\mathcal{S}_t$. From element (ii), we can uniquely determine the subset

$$\mathcal{O}^{\mathrm{done}}(\mathcal{S}_t) \subseteq \mathcal{O}$$

of operations that have already been completed by step $t$, and hence the set of remaining operations

$$\mathcal{O}^{\mathrm{rem}}(\mathcal{S}_t) := \mathcal{O} \setminus \mathcal{O}^{\mathrm{done}}(\mathcal{S}_t).$$

From elements (iii) and (iv), we know for every pair of operations in $\mathcal{O}$ their precedence relations, and for every operation in $\mathcal{O}$ the set of feasible machines together with the corresponding durations. Restricting these relations to $\mathcal{O}^{\mathrm{rem}}(\mathcal{S}_t)$ yields

(a) all remaining precedence constraints among operations in $\mathcal{O}^{\mathrm{rem}}(\mathcal{S}_t)$,

(b) all remaining duration and machine-feasibility constraints for operations in $\mathcal{O}^{\mathrm{rem}}(\mathcal{S}_t)$.

Finally, element (i) gives the current available time $AT_t^m$ of every machine $m$. From element (ii) we know exactly which operations have already been completed, and thus the set of remaining operations $\mathcal{O}^{\mathrm{rem}}(\mathcal{S}_t)$. Element (iii) specifies all precedence constraints between operations; restricting it to $\mathcal{O}^{\mathrm{rem}}(\mathcal{S}_t)$ yields the remaining O2O dependencies. Element (iv) specifies, for each operation, the set of feasible machines and their processing durations; restricting it to $\mathcal{O}^{\mathrm{rem}}(\mathcal{S}_t)$ yields the remaining O2M relations.

Any feasible completion of the schedule starting from $\mathcal{S}_t$ must therefore 1) assign each operation in $\mathcal{O}^{\mathrm{rem}}(\mathcal{S}_t)$ to exactly one machine that is feasible according to the restricted O2M graph; 2) respect all restricted precedence constraints among operations in $\mathcal{O}^{\mathrm{rem}}(\mathcal{S}_t)$; and 3) choose a start time for each operation that is not earlier than both the available time $AT_t^m$ of its assigned machine and the finish times of all its predecessors.

Let $\mathcal{F}(\mathcal{S}_t)$ denote the set of all schedules for the remaining operations that satisfy 1)–3). By construction, this feasible completion set $\mathcal{F}(\mathcal{S}_t)$ is completely determined by the tuple

$$\left( AT_t^{(\cdot)}, \mathcal{O}^{\mathrm{rem}}(\mathcal{S}_t), \text{ restricted O2O, restricted O2M} \right),$$

which itself is uniquely determined by $\mathcal{S}_t$ through elements (i)–(iv) above. In particular, $\mathcal{F}(\mathcal{S}_t)$ depends only on the current state $\mathcal{S}_t$ and not on which trajectory has led to $\mathcal{S}_t$.

Now consider any two scheduling trajectories $\tau_1$ and $\tau_2$ (possibly defined on different execution histories) that reach the same state $\mathcal{S}_t$. Let $\mathcal{F}(\mathcal{S}_t)$ denote the set of all feasible completions of the schedule starting from $\mathcal{S}_t$, that is, all feasible ways to schedule the remaining operations in $\mathcal{O}^{\mathrm{rem}}(\mathcal{S}_t)$. Since $\mathcal{F}(\mathcal{S}_t)$ is a function of $\mathcal{S}_t$ only, the feasible solution set for the remaining sub-problem induced by $\tau_1$ and $\tau_2$ is identical. For any objective that depends only on the operation finish times (for example, the makespan), each completion in $\mathcal{F}(\mathcal{S}_t)$ has the same objective value regardless of whether it is viewed as a continuation of $\tau_1$ or $\tau_2$. Hence the optimal objective value and the set of optimal completions from $\mathcal{S}_t$ are the same for both trajectories. This proves that the corresponding remaining subproblems share an identical feasible solution set and the same set of optimal solutions, which establishes the proposition. □

## E   RESCHED WITH PROXIMAL POLICY OPTIMIZATION

In the main paper we train RESCHED with REINFORCE, following standard practice in Transformer-based neural combinatorial optimization (e.g., AM (Kool et al., 2019) and POMO (Kwon et al., 2020)). This choice keeps the implementation simple and allows us to focus on our main contributions, namely the state representation and network architecture. To verify that our framework is not tied to REINFORCE, and to align with the PPO-based baseline DANIEL, we also implement a PPO version of RESCHED (denoted as RESCHED-PPO).

### E.1   TRAINING ALGORITHM WITH PPO

We also train RESCHED with Proximal Policy Optimization (PPO) using a clipped surrogate objective and generalized advantage estimation (GAE); the procedure is summarized in Algorithm 2.

### E.2   PPO CONFIGURATION AND EXPERIMENTAL SETUP

In the PPO version, we keep the state representation and network architecture identical to the RE-INFORCE version. The PPO implementation follows the clipped-surrogate variant with generalized advantage estimation (GAE). Unless otherwise stated, we adopt the same critic architecture and PPO hyperparameters as DANIEL (Wang et al., 2024b), i.e., discount factor $\gamma = 1$, GAE parameter $\lambda = 0.98$, clip range $\varepsilon = 0.2$, value-loss coefficient $c_v = 0.5$ and entropy coefficient $c_e = 0.01$. The critic shares the encoder with the policy network and adds a small MLP head that outputs a scalar state value $V_\theta(s_t)$.

**Experimental setting.**   We evaluate RESCHED-PPO on our main task, FJSP, and for simplicity we restrict the comparison to the strongest DRL-based method specifically designed for FJSP, DANIEL. We keep the same training and test splits as in the main paper: models are trained on synthetic instances from the $SD_1/SD_2$ datasets and evaluated under a greedy decoding strategy on (i) in-distribution synthetic instances, (ii) out-of-distribution synthetic instances of larger sizes, and (iii) open FJSP benchmarks.

**Training budget.**   For clarity, we measure the training budget in terms of the effective number of trajectory updates.

In vanilla REINFORCE, each sampled trajectory is used once for a single gradient update. In the main-paper configuration, we run 2000 updates and collect 1000 trajectories per update, which yields $2000 \times 1000 = 2{,}000{,}000$ trajectories. We adopted this relatively large budget to compensate for the higher variance and lower sample efficiency of REINFORCE and to stabilize training.

In PPO, for each policy update we collect a batch of $B$ trajectories from the environment and reuse them for $K$ optimization epochs, which corresponds to $BK$ effective trajectories per policy update. To disentangle the effect of the RL algorithm from that of the training budget, and to enable a fair comparison with DANIEL, we consider two budget regimes: In the ***small-budget*** setting, we choose $U = 400$ updates with $B = 50$ trajectories per update and $K = 4$ PPO epochs, resulting in $400 \times 50 \times 4 = 80{,}000$ effective trajectory updates. This matches the training budget used

---

**Algorithm 2** Training RESCHED with PPO

---

1: **Input:** scheduling environment $\mathcal{E}$, model parameters $\theta$, total number of policy updates $U$, number of trajectories per update $B$, mini-batch size $M$, number of PPO epochs $K$ per update, discount factor $\gamma$, GAE parameter $\lambda$, clip range $\varepsilon$, learning rate $\alpha$
2: **for** update $u = 1$ to $U$ **do**
3:     Set old parameters $\theta_{\text{old}} \leftarrow \theta$;
4:     Initialize buffer $\mathcal{D} \leftarrow \emptyset$;
5:     **for** $i = 1$ to $B$ **do**
6:         Sample a training instance from $\mathcal{E}$ and reset to get initial state $s_0$;
7:         Initialize trajectory $\mathcal{T} \leftarrow \emptyset$;
8:         **repeat**
9:             Encode current state $s_t$ and compute $\pi_{\theta_{\text{old}}}(\cdot \mid s_t)$ and $V_{\theta_{\text{old}}}(s_t)$;
10:             Sample action $a_t \sim \pi_{\theta_{\text{old}}}(\cdot \mid s_t)$;
11:             Execute $a_t$ to obtain reward $r_t$ and next state $s_{t+1}$;
12:             Store $(s_t, a_t, r_t, V_{\theta_{\text{old}}}(s_t), \log \pi_{\theta_{\text{old}}}(a_t \mid s_t))$ into $\mathcal{T}$;
13:             $s_t \leftarrow s_{t+1}$;
14:         **until** the scheduling instance is finished
15:         Append all time steps in $\mathcal{T}$ to buffer $\mathcal{D}$;
16:     **end for**
17:     Using rewards and old values in $\mathcal{D}$, compute returns $R_t$ and advantages $\hat{A}_t$ with GAE$(\gamma, \lambda)$;
18:     **for** PPO epoch $k = 1$ to $K$ **do**
19:         Shuffle $\mathcal{D}$ and split into mini-batches of size $M$;
20:         **for** each mini-batch $\mathcal{B} \subset \mathcal{D}$ **do**
21:             For all $(s_t, a_t)$ in $\mathcal{B}$, compute $\pi_\theta(\cdot \mid s_t)$ and $V_\theta(s_t)$;
22:             Let $p_t = \pi_\theta(a_t \mid s_t)$, $p_t^{\text{old}} = \pi_{\theta_{\text{old}}}(a_t \mid s_t)$, and $r_t = p_t / p_t^{\text{old}}$;
23:             Policy loss:

$$\mathcal{L}_{\text{policy}} = -\mathbb{E}_{t \in \mathcal{B}}\Big[\min\big(r_t \hat{A}_t, \, \text{clip}(r_t, 1 - \varepsilon, 1 + \varepsilon)\hat{A}_t\big)\Big];$$

24:             Value loss with clipping:

$$\mathcal{L}_{\text{value}} = \mathbb{E}_{t \in \mathcal{B}}\Big[\max\Big((V_\theta(s_t) - R_t)^2,$$
$$(V_{\theta_{\text{old}}}(s_t) + \text{clip}(V_\theta(s_t) - V_{\theta_{\text{old}}}(s_t), -\varepsilon, \varepsilon) - R_t)^2\Big)\Big].$$

25:             Entropy bonus: $\mathcal{L}_{\text{entropy}} = -\mathbb{E}_{t \in \mathcal{B}}[\mathcal{H}(\pi_\theta(\cdot \mid s_t))]$;
26:             Total loss: $\mathcal{L} = \mathcal{L}_{\text{policy}} + c_v \mathcal{L}_{\text{value}} + c_e \mathcal{L}_{\text{entropy}}$;
27:             Update parameters: $\theta \leftarrow \theta - \alpha \nabla_\theta \mathcal{L}$;
28:         **end for**
29:     **end for**
30:     Optionally step the learning-rate scheduler and evaluate $\pi_\theta$ on a validation set;
31: **end for**
32: **Output:** trained model parameters $\theta$

---

by DANIEL, for which we report the original DANIEL results while retraining both RESCHED-REINFORCE and RESCHED-PPO under the same budget. In the ***large-budget*** setting, we increase $U$ by a factor of 25 so that both DANIEL and RESCHED-PPO are retrained with approximately 2,000,000 effective trajectory updates, matching the budget of our original RESCHED-REINFORCE configuration, for which we directly reuse the main-paper model.

### E.3   COMPARISON OF RESCHED AND DANIEL UNDER DIFFERENT TRAINING BUDGETS

The results for DANIEL, RESCHED-REINFORCE, and RESCHED-PPO are reported in Table 10.

**In-distribution performance under a small training budget.** Under the small-budget configuration, RESCHED-PPO converges in noticeably fewer updates than RESCHED-REINFORCE and

Table 10: Results on FJSP: in-distribution (top); out-of-distribution (middle); benchmark (bottom)

| Dataset | Size | | PDRs | | | | DANIEL | Small training budget | | DANIEL | Large training budget | | OR-Tools[1] |
|---|---|---|---|---|---|---|---|---|---|---|---|---|---|
| | | | FIFO | SPT | MOPNR | MWKR | | RESCHED-REINFORCE | RESCHED-PPO | | RESCHED-REINFORCE | RESCHED-PPO | |
| SD₁ | 10×5 | Gap(%)↓ | 24.06 | 34.76 | 19.87 | 17.58 | **10.87** | 14.61 | 11.20 | **9.22** | 12.25 | 11.48 | 96.32 (5%) |
| | 20×5 | Gap(%)↓ | 14.87 | 22.56 | 13.85 | 11.51 | **5.03** | 8.51 | 5.84 | **3.08** | 4.63 | 4.20 | 188.15 (0%) |
| | 15×10 | Gap(%)↓ | 28.65 | 38.22 | 20.68 | 19.41 | 12.42 | 12.91 | **9.41** | 10.84 | 6.51 | **5.21** | 143.53 (7%) |
| | 20×10 | Gap(%)↓ | 19.22 | 30.25 | 12.20 | 10.30 | **1.31** | 6.97 | 3.50 | **-0.43** | 0.48 | -0.36 | 195.98 (0%) |
| SD₂ | 10×5 | Gap(%)↓ | 76.47 | 57.96 | 72.52 | 70.01 | 25.68 | 19.06 | **15.77** | 24.75 | 16.36 | **14.24** | 326.24 (96%) |
| | 20×5 | Gap(%)↓ | 74.59 | 38.91 | 74.58 | 71.31 | 11.52 | 10.76 | **9.21** | 8.86 | 9.87 | **6.83** | 602.04 (0%) |
| | 15×10 | Gap(%)↓ | 132.23 | 86.74 | 125.32 | 121.45 | 57.16 | **24.03** | 25.02 | 53.94 | 18.14 | **16.73** | 377.17 (28%) |
| | 20×10 | Gap(%)↓ | 135.27 | 78.82 | 129.09 | 124.98 | 31.58 | **17.91** | 19.38 | 28.89 | 14.18 | **13.79** | 464.16 (1%) |
| Avg. | | Gap(%)↓ | 63.17 | 48.53 | 58.51 | 55.82 | 19.45 | 14.35 | **12.42** | 17.39 | 10.30 | **9.02** | - |

| Dataset | Size | | Top PDRs | | DANIEL | | Small training budget | | | | DANIEL | | Large training budget | | | | OR-Tools |
|---|---|---|---|---|---|---|---|---|---|---|---|---|---|---|---|---|---|
| | | | | | | | RESCHED-REINFORCE | | RESCHED-PPO | | | | RESCHED-REINFORCE | | RESCHED-PPO | | |
| | | | SPT | MWKR | 10×5 | 20×10 | 10×5 | 20×10 | 10×5 | 20×10 | 10×5 | 20×10 | 10×5 | 20×10 | 10×5 | 20×10 | |
| SD₁ | 30×10 | Gap(%)↓ | 27.47 | 13.96 | 5.10 | **2.50** | 9.26 | 4.43 | 3.21 | 4.50 | 2.05 | **1.45** | 3.44 | 2.69 | 2.81 | 1.91 | 274.67 (6%) |
| | 40×10 | Gap(%)↓ | 21.66 | 13.37 | 3.65 | **1.52** | 8.04 | 3.37 | 2.19 | 3.23 | 0.98 | **0.53** | 2.54 | 1.64 | 2.06 | 1.45 | 365.96 (3%) |
| SD₂ | 30×10 | Gap(%)↓ | 59.74 | 122.89 | 14.85 | 11.95 | 37.76 | **8.17** | 8.97 | 11.96 | 21.51 | 18.59 | 8.79 | 6.30 | **5.16** | 7.05 | 692.26 (0%) |
| | 40×10 | Gap(%)↓ | 38.74 | 108.66 | 0.52 | -1.67 | 23.25 | -3.39 | **-3.51** | -1.29 | 4.60 | 0.05 | -2.40 | -4.58 | **-6.02** | **-6.02** | 998.39 (0%) |
| Avg. | | Gap(%)↓ | 36.90 | 64.72 | 6.03 | 3.58 | 19.58 | 3.15 | **2.72** | 4.60 | 7.29 | 5.16 | 3.09 | 1.51 | **1.00** | 1.10 | - |

| Strategy | Dataset | | MWKR (Top PDR) | DANIEL | | RESCHED-REINFORCE | | RESCHED-PPO | | 2SGA | OR-Tools | UB[2] |
|---|---|---|---|---|---|---|---|---|---|---|---|---|
| | | | | 10×5 | 15×10 | 10×5 | 15×10 | 10×5 | 15×10 | | | |
| **Small training budget** | Brandimarte | Gap(%)↓ | 28.91 | 13.58 | 12.97 | 13.50 | 14.33 | **10.52** | 10.77 | 175.20(3.17%) | 174.20(1.5%) | 172.7 |
| | Hurink(edata) | Gap(%)↓ | 18.60 | 16.33 | **14.41** | 18.06 | 16.34 | 18.29 | 20.00 | - | 1028.93(-0.03%) | 1028.88 |
| | Hurink(rdata) | Gap(%)↓ | 13.86 | 11.42 | 12.07 | 10.28 | **9.92** | 10.45 | 13.44 | - | 935.80(0.11%) | 934.28 |
| | Hurink(vdata) | Gap(%)↓ | 4.22 | 3.28 | 3.75 | **2.67** | 2.82 | 4.26 | 3.60 | 812.20(0.39%) | 919.60(-0.01%) | 919.50 |
| | Avg. | Gap(%)↓ | 16.40 | 11.15 | **10.80** | 11.13 | 10.86 | 11.95 | | - | - | - |
| **Large training budget** | Brandimarte | Gap(%)↓ | 28.91 | 14.10 | 14.58 | **9.08** | 12.49 | 10.34 | 10.97 | 175.20(3.17%) | 174.20(1.5%) | 172.7 |
| | Hurink(edata) | Gap(%)↓ | 18.60 | **14.67** | 15.71 | 15.48 | 16.34 | 16.25 | 18.16 | - | 1028.93(-0.03%) | 1028.88 |
| | Hurink(rdata) | Gap(%)↓ | 13.86 | 11.20 | 10.34 | **10.18** | 10.31 | 10.59 | 10.42 | - | 935.80(0.11%) | 934.28 |
| | Hurink(vdata) | Gap(%)↓ | 4.22 | 3.17 | 3.42 | 3.48 | **2.55** | 6.41 | 3.98 | 812.20(0.39%) | 919.60(-0.01%) | 919.50 |
| | Avg. | Gap(%)↓ | 16.40 | 10.79 | 11.01 | **9.56** | 10.42 | 10.90 | 10.88 | - | - | - |

1. OR-Tools (1800s per instance): solution and optimal ratio reported;
2. UB is the best-known solution (Behnke & Geiger, 2012), used as the baseline to compute gaps;
3. **Instance-wise average gap** is reported to reduce bias from varying instance scales.

achieves the best average in-distribution performance, with an average gap of 12.42% compared to 14.35% for RESCHED-REINFORCE and 19.45% for DANIEL. Even the REINFORCE version, despite its slower convergence, still surpasses DANIEL in terms of average optimality gap, indicating that the main gain comes from our architecture rather than from using a larger training budget. PPO further exploits the limited data more efficiently than REINFORCE, yielding the best average results among all compared methods.

**Out-of-distribution performance under a small training budget.** In the out-of-distribution setting, models trained on SD₁-10×5 and SD₁-20×10 are evaluated on larger unseen instances. Under the small training budget, as shown in Table 10, RESCHED-PPO trained only on SD₁-10×5 already achieves the best average OOD performance, achieves the best *average* OOD performance (2.72% gap), outperforming both DANIEL and RESCHED-REINFORCE, showing that our architecture combined with PPO can generalize well even when trained on the smallest problem size with limited data. In contrast, the REINFORCE version trained on SD₁-10×5 generalizes poorly under this small budget, which is attributed to its lower sample efficiency. However, when the training size is increased to SD₁-20×10 (with the same budget), its OOD performance improves substantially and becomes comparable to DANIEL. Overall, these results indicate that our architecture does generalize beyond the training size, with PPO exploiting limited data more efficiently (as PPO trains the policy multiple times on the same data), whereas REINFORCE requires somewhat richer training instances to reach a similar level of OOD performance.

**Open benchmark performance under a small training budget.** On the open benchmark sets, all DRL-based methods achieve very similar average performance under the small training budget. In particular, the REINFORCE and PPO versions of RESCHED trained on small synthetic instances remain competitive with the best DANIEL configuration. Specifically, the REINFORCE-based RESCHED requires approximately 30 minutes to train on the SD₁-10×5 instances, while the PPO-based version trains in about 20 minutes. In comparison, HGNN (Song et al., 2023) and DANIEL (Wang et al., 2024b) require about 21 minutes and 18 minutes, respectively, to complete the same training. This indicates that, with only a small amount of training data and a short training time , current DRL-based methods already reach a reasonably strong level on real-world scheduling benchmarks, highlighting their practical potential for real scheduling applications.

**In-distribution performance under a large training budget.** Under the large-budget setting, both RESCHED versions benefit from the increased data: the average gap of RESCHED-REINFORCE drops from 14.35% to 10.30%, and RESCHED-PPO further improves it to 9.02%. RESCHED-PPO consistently improves over the REINFORCE version across almost all datasets. DANIEL also benefits from the larger budget, yet even the REINFORCE version of RESCHED now clearly outperforms DANIEL on most SD$_2$ and medium-sized SD$_1$ instances, and the RESCHED-PPO achieves the best average in-distribution performance overall. This shows that, even when DANIEL is given a comparable large training budget, the proposed architecture (especially with PPO) remains substantially stronger.

**Out-of-distribution performance under a large training budget.** In the out-of-distribution setting with a large training budget, both RESCHED versions clearly outperform DANIEL on average, and the PPO version consistently improves over the REINFORCE version across most datasets.

Interestingly, when we increase the training budget of DANIEL by 25×, its OOD performance on the SD$_2$ datasets does not improve and even degrades compared to the original setting. This suggests that, under the current training setup, DANIEL does not clearly benefit from additional data, which may partly explain why the original work chose a relatively small training budget. In contrast, RESCHED continues to improve when the training budget is increased, indicating that our architecture can effectively leverage more trajectories.

**Open benchmark performance under a large training budget.** On the open benchmark, all DRL-based methods achieve very similar performance under the large training budget, with only minor differences in gaps. RESCHED (both REINFORCE and PPO) remains competitive with DANIEL, indicating that models trained on synthetic SD$_1$ instances transfer reasonably well to these open benchmark and do not exhibit noticeable overfitting to the synthetic distribution.

These new experiments show that our architecture works well with both REINFORCE and PPO: the PPO version converges faster and further improves in-distribution and out-of-distribution performance, while the REINFORCE version already remains competitive or better than DANIEL under matched budgets. **In particular, replacing REINFORCE with the stronger PPO algorithm further improves RESCHED's performance, demonstrating that our framework can directly benefit from stronger RL algorithms.** Overall, across all budgets and RL algorithms, RESCHED consistently matches or outperforms DANIEL, indicating that the benefits come from our framework rather than from the large amount of training data.

## F ADDITIONAL ABLATIONS AND DESIGN CHOICES OF RESCHED

**Reward Design** In our early framework explorations, we tested three different reward formulations: $\Delta$-makespan (the change in makespan before and after taking an action), estimated mean reward Song et al. (2023), and estimated lower-bound reward Zhang et al. (2020); Wang et al. (2024b). As summarized in Table 12, all three variants are able to train a reasonable policy, but the choice of reward has a noticeable impact on generalization. In particular, the delta-makespan reward tends to fit the synthetic distribution more aggressively: it can slightly improve average performance on synthetic OOD instances, but it significantly degrades performance on the open benchmarks compared to the estimated lower-bound reward, which we interpret as a form of overfitting to the synthetic training distribution. As the estimated lower-bound reward is commonly used in DRL based scheduling methods like L2D (Zhang et al., 2020) and DANIEL (Wang et al., 2024b), we also use it to align with the standard practice.

**Discount Factor $\gamma$** We initially set the discount factor to $\gamma = 0.99$, following standard deep RL practices. To investigate the impact of this hyperparameter, we also performed additional experiments with $\gamma = 1.0$. The results, summarized in Table 13, show that our method is not sensitive to this hyperparameter choice, and we have maintained $\gamma = 0.99$ in our main configuration.

Table 12: Effect of different reward on the performance of RESCHED.

| Dataset | Est LB | Est Mean | $\Delta$-Makespan |
|---|---|---|---|
| **In-distribution** | | | |
| SD$_1$-10×05 | **12.25** | 12.56 | 13.05 |
| **Out-of-distribution** | | | |
| SD$_1$-30×10 | 3.44 | **3.29** | 3.30 |
| SD$_1$-40×10 | 2.54 | **2.15** | 2.41 |
| SD$_2$-30×10 | 8.79 | 9.19 | **7.53** |
| SD$_2$-40×10 | -2.40 | -2.57 | **-4.30** |
| Avg. | 3.09 | 3.015 | **2.24** |
| **Open Benchmark** | | | |
| Brandimarte | **9.08** | 9.66 | 15.35 |
| Hurink (edata) | **15.48** | 16.58 | 17.27 |
| Hurink (rdata) | **10.18** | 10.34 | 11.62 |
| Hurink (vdata) | 3.48 | **2.75** | 4.50 |
| Avg. | **9.56** | 9.83 | 12.19 |

Table 13: Effect of the discount factor $\gamma$ on the performance of RESCHED.

| Dataset | RESCHED-0.99-Avg. | RESCHED-1.0-Avg. | RESCHED-1.0-seed1 | RESCHED-1.0-seed2 | RESCHED-1.0-seed3 |
|---|---|---|---|---|---|
| **In-distribution** | | | | | |
| SD$_1$-10×05 | **11.75** | **11.75** | 11.41 | 11.76 | 12.07 |
| **Out-of-distribution** | | | | | |
| SD$_1$-30×10 | **3.10** | **3.10** | 2.63 | 3.69 | 2.97 |
| SD$_1$-40×10 | 2.23 | **2.19** | 2.00 | 2.51 | 2.06 |
| SD$_2$-30×10 | **9.04** | 10.03 | 8.28 | 11.58 | 10.22 |
| SD$_2$-40×10 | **-2.61** | -2.09 | -3.55 | -0.11 | -2.61 |
| Avg. | **2.94** | 3.31 | 2.34 | 4.42 | 3.16 |
| **Open Benchmark** | | | | | |
| Brandimarte | **10.28** | 11.52 | 12.44 | 10.32 | 11.80 |
| Hurink (edata) | **15.84** | 16.26 | 16.03 | 16.89 | 15.85 |
| Hurink (rdata) | **10.13** | 10.50 | 10.49 | 10.11 | 10.91 |
| Hurink (vdata) | 2.90 | **2.67** | 2.66 | 2.37 | 2.98 |
| Avg. | **9.78** | 10.24 | 10.41 | 9.92 | 10.39 |

Values are average optimality gaps (%) w.r.t. the best-known upper bounds.
"-Avg." denote averages over four and three seeds, respectively.

## G FREQUENTLY ASKED QUESTIONS

### G.1 RESCHED EXCEL ON FJSP BUT ARE LESS COMPETITIVE ON RECENT JSP BASELINES. WHAT STRUCTURAL OR DISTRIBUTIONAL DIFFERENCES MAKE JSSP HARDER FOR RESCHED?

As the instance distribution of FJSP moves closer to a pure JSSP (one feasible machine per operation), the relative advantage of FJSP-oriented DRL methods becomes smaller. This phenomenon is not specific to RESCHED; it also appears on existing FJSP baselines. For example, on the Hurink benchmarks for FJSP, edata has on average only 1.15 feasible machines per operation (very close to JSSP), whereas rdata has 2 and vdata has about (m/2) machines per operation. As shown in Table 14, all three FJSP methods (HGNN, DANIEL, and RESCHED) perform worst on edata and improve as the flexibility increases from edata → rdata → vdata.

Beyond the empirical trend observed on the Hurink dataset, we would like to highlight a structural reason for this behavior. Methods for the Flexible JSP (FJSP), including our own, are explicitly designed to model the interactions between operations and machines. Our architecture, for instance, incorporates a dedicated O2M branch and machine nodes, whose primary function is to resolve assignments among multiple feasible machines and balance their loads.

In strict JSP instances, however, this flexibility is absent, as each operation is assigned to a single feasible machine. Consequently, the O2M decision structure becomes less critical, and a portion of the model's dedicated capacity is consequently less informative. The challenge in these instances

Table 14: Perfprmance degradation analysis for RL approaches.

| Dataset | Avg. machines/oper | HGNN 10×5 | DANIEL 10×5 | ReSched 10×5 |
|---|---|---|---|---|
| Hurink (edata) | 1.15 | 15.53 | 16.33 | 15.48 |
| Hurink (rdata) | 2 | 11.15 | 11.42 | 10.18 |
| Hurink (vdata) | $m/2$ | 4.25 | 3.28 | 3.48 |

shifts predominantly to resolving fine-grained sequencing conflicts along predetermined machine routes.

In contrast, many recent JSP-tailored methods simplify the representation by pooling machine-related information into aggregated operation features, effectively modeling the problem on an operation-only graph (Lee & Kim, 2024; Zhang et al., 2020; Park et al., 2021b; Chen et al., 2023; Park et al., 2021a; Iklassov et al., 2023). This abstraction can enable the model to concentrate more directly on resolving sequencing conflicts within a fixed disjunctive graph, which may yield stronger performance on classic JSP benchmarks.

However, we posit that in practical manufacturing settings, an operation is intrinsically linked to its machine resources, as embodied by attributes like machine-specific processing times. This is why our framework intentionally maintains distinct operation and machine nodes with explicit O2O and O2M edges. This architectural choice necessarily trades off some degree of short-term, JSP-specific optimality to achieve a more generalizable, faithful, and extensible model. It is specifically designed to accommodate FJSP and other complex variants where machine flexibility and richer precedence structures are fundamental to the problem.

### G.2 THE FJSP, JSSP, AND FFSP REPRESENT RELATIVELY LIMITED AND LESS COMPLEX SCHEDULING PROBLEMS. HOW COULD THIS APPROACH BE EXTENDED TO HANDLE A BROADER RANGE OF SCHEDULING PROBLEMS, INCLUDING THOSE WITH MORE DIVERSE AND REALISTIC CONSTRAINTS ENCOUNTERED IN REAL-WORLD APPLICATIONS?

The goal of RESCHED is precisely to build a simple and generic framework that can be quickly adapted to different scheduling or resource-allocation scenarios. To this end, we start from FJSP as a generic formulation, view other classic shceudling problems (JSSP, FFSP) through a heterogeneous operation–machine graph, and deliberately avoid problem-specific features or heuristics in both the state representation and the network architecture. This is why the same model can already handle FJSP, JSSP, and FFSP without any architectural changes.

As a concrete example, dependent task offloading in mobile edge computing (Wang et al., 2022; Xiao et al., 2025; Wang et al., 2020) can be viewed as an instance of our formulation. In this problem, an application is modeled as a DAG of computation tasks with precedence constraints, and the scheduler must decide for each ready task whether to execute it locally or offload it to one of several heterogeneous edge/cloud servers, subject to communication and resource limits, in order to minimize end-to-end latency (or a latency–energy trade-off).

In this case, each computation task in DAG corresponds to an operation node, and each local device, edge server, or cloud node corresponds to a machine node. Task dependencies form the O2O edges, while the end-to-end latency for executing a task on a given resource (communication plus computation, or purely computation for local execution) defines the O2M edge durations, with infeasible assignments masked out. Under this mapping, the task DAG is directly represented as the O2O connection mask fed into the operation branch, so no architectural change is required. The scheduler's decision at each step is to select a feasible task–resource pair, and the objective is typically a function of task completion times such as total latency or a latency–energy trade-off. Our state representation and dual-branch Transformer can thus be reused without modification; only the duration model and reward definition need to be adapted, illustrating how RESCHED can be extended to more realistic scheduling scenarios.

## H STATEMENT ON THE USE OF LARGE LANGUAGE MODELS (LLMS)

We used large language models (e.g., ChatGPT) as a general-purpose assistant for language polishing (grammar, wording, clarity) and for suggesting occasional non-substantive code snippets. LLMs

were not used for problem formulation, algorithm/model design, experimental design or analysis, data generation, or drawing conclusions. All core code and technical content were implemented and verified by us. We reviewed and edited all LLM-assisted text and code, and take full responsibility for every part of the manuscript, including sections that benefited from LLM assistance.

