# OpenReview forum: "RESCHED: Rethinking Flexible Job Shop Scheduling from a Transformer-based Architecture with Simplified States"
_ICLR.cc/2026/Conference — ICLR 2026 Poster_

### Official Review · Reviewer_WVwe · 2025-10-23

**Soundness:** 2
**Presentation:** 3
**Contribution:** 2
**Rating:** 2
**Confidence:** 4

**Summary:**

This paper introduces ReSched that emphasizes a minimalist approach by redefining the state representation of FJSP, by only using four features compared to the current state-of-the-art method DANIEL, which uses a total of 26 features.  The paper also shows that many of the features of DANIEL are redundant and that performance increases when some are removed. Additionally, the paper introduces a novel network architecture, whereby they show through extensive testing the effect of the components of the network architecture. Their results indicate that ReSched outperforms DANIEL and other baselines, without a significant increase in the evaluation runtime.

**Strengths:**

The authors show that existing DRL work uses too many features to train, and they also show an ablation study for the features of DANIEL, further strengthening their claims for a minimalist approach.

The Ablation study of ReSched in Table 5 is well done, and it shows the effect of each introduced component of their network architecture.

**Weaknesses:**

The use of the REINFORCE algorithm over PPO is not well-justified. It is well-known that the vanilla REINFORCE method suffers from high variance, and the paper does not report training across multiple seeds. This raises doubts about the results presented in the paper, as training over multiple seeds might reveal that DANIEL outperforms ReSched. While DANIEL also trained on only a single seed, they used PPO, which is known to be more stable.

REINFORCE is also known to be highly sample-inefficient. The paper states that ReSched trains for smaller instances for 2000 epochs, with 1000 instances each epoch, with a batch size of 50. This is significantly longer than baselines, as both HGNN and DANIEL, who also uses 1000 instances, but with a batch size of 20, train in a single “epoch”, meaning they only use these instances once. This means that DANIEL and HGNN will have $1000 \times 20= 20000$ different episodes to train on, whereas ReSched has $2000 \times 50 \times 1000= 100000000$. This means that ReSched has 5000 times the training data compared to the baselines, which it compares to. This could mean that HGNN and, especially, DANIEL could outperform ReSched whenever they are trained on the same amount of data.

The contribution of ReSched is marginal in my opinion, since it only introduces a new state representation and network architecture. Moreover, due to previously stated weaknesses, I am not fully convinced about the results and, consequently, the significance of these novel parts.

The paper uses Definitions, Propositions, Corollaries, and Remarks quite liberally. For example, Corollary 1 is completely based on unproven Proposition 1. I can understand that Proposition 1 is likely true, but I would not state it in this way without giving a formal proof.

**Questions:**

Could the authors explain why $\gamma=0.99$, when it is normally set to $\gamma=1$, such as for L2D and DANIEL?

Could the authors maybe show if their proposed network architecture works well, with PPO or REINFORCE with a custom baseline? Their current motivation of REINFORCE being simpler is not strong, given that we use methods like PPO for a reason, like better sample efficiency and more stable training.

Could the authors show the performance of ReSched when trained on a similar amount of data as DANIEL and HGNN?

In Table 3, it is unclear whether ReSched uses the greedy or sampling evaluation for JSP. Could you please clarify which one is used?

---

> ### Author Response · Authors · 2025-12-01
> **REINFORCE Variance and Multi-Seed Evaluation**
>
> > **W1**: The use of the REINFORCE algorithm over PPO is not well-justified. It is well-known that the vanilla REINFORCE method suffers from high variance, and the paper does not report training across multiple seeds. This raises doubts about the results presented in the paper, as training over multiple seeds might reveal that DANIEL outperforms ReSched. While DANIEL also trained on only a single seed, they used PPO, which is known to be more stable.
>
> We thank the reviewer for raising this concern. Our main contribution lies in the state representation and network architecture rather than in proposing a new RL algorithm. In the initial version, we chose a REINFORCE-style policy gradient mainly because it is simple to implement and follows the standard practice in Transformer-based neural combinatorial optimization, such as the AM[AR10] and POMO[AR11], where vanilla REINFORCE is commonly used to train the policy network. We agree, however, that vanilla REINFORCE is known to have higher variance than PPO. For this reason, in the revised manuscript (Sec. 4.2.2) we now discuss this trade-off and clarify our motivation for using REINFORCE in the initial version.
>
> To further address this concern, we reran ReSched on the SD1-10x5 training setting with three additional independent runs using different random seeds. For each run, we evaluated the policy across in-distribution, out-of-distribution, and open benchmark test settings to comprehensively test the performance of ReSched. For ease of comparison, Table RT16 reports in the second column the average over all four independent runs (the originally reported run plus the three new ones), while the subsequent columns list the results of each individual run. On out-of-distribution and open benchmark tests, all ReSched runs outperform DANIEL, while in-distribution performance on SD1–10×5 remains comparable. Moreover, the variation across seeds is much smaller than the performance gap between ReSched and DANIEL, indicating that our gains are robust and not due to a fortunate choice of random seed.
>
> **Table RT16. Performance of ReSched trained with REINFORCE across four random seeds on SD1–10×5, compared with DANIEL. Values are average optimality gaps (%) w.r.t. the best-known upper bounds. “ReSched-Avg.” denotes the mean over all four ReSched runs (original + three additional seeds).**
> |In-distribution|DANIEL (Origin Reported)|ReSched-Avg.|ReSched-REINFORCE (Origin Reported)|ReSched-REINFORCE-seed1|ReSched-REINFORCE-seed2|ReSched-REINFORCE-seed3|
> |-|-|-|-|-|-|-|
> |SD1-10x05|**10.87**|11.75|12.25|11.40|12.33|11.03|
> |**Out-of-distribution**|**DANIEL (Origin Reported)**|**ReSched-Avg.**|**ReSched-REINFORCE (Origin Reported)**|**ReSched-REINFORCE-seed1**|**ReSched-REINFORCE-seed2**|**ReSched-REINFORCE-seed2**|
> |SD1-30x10|5.10|**3.10**|3.44|2.46|3.39|3.09|
> |SD1-40x10|3.65|**2.23**|2.54|1.64|2.46|2.26|
> |SD2-30x10|14.85|**9.04**|8.79|9.35|9.78|8.22|
> |SD2-40x10|0.52|**-2.61**|-2.40|-3.00|-1.80|-3.23|
> |Avg.|6.03|**2.94**|3.09|2.61|3.46|2.59|
> |**Open Benchmark**|**DANIEL (Origin Reported)**|**ReSched-Avg.**|**ReSched-REINFORCE (Origin Reported)**|**ReSched-REINFORCE-seed1**|**ReSched-REINFORCE-seed2**|**ReSched-REINFORCE-seed2**|
> |Brandimarte|13.58|**10.28**|9.08|11.70|10.10|10.24|
> |Hurink(edata)|16.33|**15.84**|15.48|16.46|15.52|15.88|
> |Hurink(rdata)|11.42|**10.13**|10.18|10.30|10.46|9.57|
> |Hurink(vdata)|3.28|**2.90**|3.48|2.60|2.55|2.95|
> |Avg.|11.15|**9.78**|9.56|10.27|9.66|9.66|
>
> [AR10]. Kool W, Van Hoof H, Welling M. Attention, learn to solve routing problems!. International Conference on Learning Representations, 2019.
> [AR11]. Kwon Y D, Choo J, Kim B, et al. Pomo: Policy optimization with multiple optima for reinforcement learning. Advances in Neural Information Processing Systems, 2020.

---

> ### Author Response · Authors · 2025-12-01
> **ReSched with PPO (part1/3)**
>
> > **W2**: REINFORCE is also known to be highly sample-inefficient. The paper states that ReSched trains for smaller instances for 2000 epochs, with 1000 instances each epoch, with a batch size of 50. This is significantly longer than baselines, as both HGNN and DANIEL, who also uses 1000 instances, but with a batch size of 20, train in a single “epoch”, meaning they only use these instances once. This means that DANIEL and HGNN will have 1000x20 = 20000 different episodes to train on, whereas ReSched has 2000x50x1000 = 2000000. This means that ReSched has 5000 times the training data compared to the baselines, which it compares to. This could mean that HGNN and, especially, DANIEL could outperform ReSched whenever they are trained on the same amount of data.
>
> > **Q2**: Could the authors maybe show if their proposed network architecture works well, with PPO or REINFORCE with a custom baseline? Their current motivation of REINFORCE being simpler is not strong, given that we use methods like PPO for a reason, like better sample efficiency and more stable training.
>
> > **Q3**: Could the authors show the performance of ReSched when trained on a similar amount of data as DANIEL and HGNN?
>
> We agree that our original training configuration uses more environment interactions than DANIEL and HGNN, and vanilla REINFORCE indeed typically requires more samples to converge stably.
>
> However, the “5000× more data” figure in the review comes from a misunderstanding of our setup. In our implementation, each epoch consists of 1000 instances and we generate one trajectory per instance, so 2000 epochs correspond to 2000×1000=2,000,000 trajectories (convergence is usually reached after about 1,000,000 trajectories, and we extended the budget to stabilize the results). By contrast, DANIEL generates 1000 instances with a batch size of 20 once, yielding 1000×20=20,000 trajectories; with PPO reusing each collected trajectory for K optimization epochs (with K=4 in DANIEL), the effective number of trajectory updates is 1000×20×4=80,000. Thus, while ReSched does use a larger training budget, the gap is about 80,000 (**roughly 25×**), rather than 5000×.
>
> To further address the reviewer’s concerns about sample efficiency and fairness (W2, Q2, Q3), we conducted two additional sets of experiments with both REINFORCE and a **newly implemented PPO-based version** of ReSched.
>
> To further investigate whether this larger budget is responsible for our performance gains, we conducted additional experiments under two training settings: a **small-budget** setting matched to DANIEL, and a **large-budget setting** comparable to our original configuration. For a comprehensive assessment, we follow the same evaluation protocol as in the main paper and evaluate under a greedy strategy on in-distribution, out-of-distribution, and open benchmark settings.
>
> These new experiments show that our architecture works well with both REINFORCE and PPO: the PPO version converges faster and further improves in-distribution and out-of-distribution performance, while the REINFORCE version already remains competitive or better than DANIEL under matched budgets. **In particular, replacing REINFORCE with the stronger PPO algorithm further improves ReSched’s performance, demonstrating that our framework can directly benefit from stronger RL algorithms.** Overall, across all budgets and RL algorithms, ReSched consistently matches or outperforms DANIEL, indicating that the benefits come from our framework rather than from the large amount of training data.

---

> ### Author Response · Authors · 2025-12-01
> **ReSched with PPO (part2/3)**
>
> (1). **Small-budget setting**
> In the small-budget setting, we retrain the REINFORCE-based ReSched with all other hyperparameters unchanged, while reducing the number of epochs so that the total number of trajectories is 80 × 1000 = 80,000, which is on the same order as DANIEL. For the PPO version of ReSched, we use only 400 epochs with a batch size of 50, and reuse each collected batch for K = 4 PPO updates (the same K as in DANIEL), again resulting in 400 × 50 × 4 = 80,000 effective trajectory updates. The PPO version of ReSched adopts exactly the same critic architecture and PPO hyperparameters as DANIEL.
>
> Under this small training budget, the PPO version converges faster and achieves better in-distribution performance than both DANIEL and the REINFORCE version of ReSched. Even the REINFORCE version, despite its slower convergence, still surpasses DANIEL in terms of average optimality gap, indicating that the main gain comes from our architecture rather than from using a larger training budget. PPO further exploits the limited data more efficiently than REINFORCE, yielding the best average results among all compared methods.
>
> **Table RT6. In-distribution performance of DANIEL and ReSched trained with REINFORCE vs. PPO under a small training budget. Values are average optimality gaps (%) w.r.t. the best-known upper bounds.**
> |In-distribution|DANIEL|ReSched-REINFORCE|ReSched-PPO|
> |-|-|-|-|
> |SD1-10x05|**10.87**|14.61|11.20|
> |SD1-20x05|**5.03**|8.51|5.84|
> |SD1-15x10|12.42|12.91|**9.41**|
> |SD1-20x10|**1.31**|6.97|3.50|
> |SD2-10x05|25.68|19.06|**15.77**|
> |SD2-20x05|11.52|10.76|**9.21**|
> |SD2-15x10|57.16|**24.03**|25.015|
> |SD2-20x10|31.58|**17.913**|19.38|
> |Avg.|19.45|14.35|**12.42**|
>
>
> In the out-of-distribution setting, models trained on SD1-10×5 and SD1-20×10 are evaluated on larger unseen instances. Under the small training budget, the PPO version trained only on SD1-10×5 already achieves the best average OOD performance, showing that our architecture combined with PPO can generalize well even when trained on the smallest problem size with limited data. In contrast, the REINFORCE version trained on SD1-10×5 generalizes poorly under this small budget, which is attributed to its lower sample efficiency. However, when the training size is increased to SD1-20×10 (with the same budget), its OOD performance improves substantially and becomes comparable to DANIEL. Overall, these results indicate that our architecture does generalize beyond the training size, with PPO exploiting limited data more efficiently (as PPO trains the policy multiple times on the same data), whereas REINFORCE requires somewhat richer training instances to reach a similar level of OOD performance.
>
> **Table RT7. Out-of-distribution generalization of DANIEL and ReSched trained with REINFORCE vs. PPO under a small training budget. Values are average optimality gaps (%) w.r.t. the best-known upper bounds.**
> |Out-of-distribution|DANIEL-SD1-10x5|DANIEL-SD1-20x10|ReSched-REINFORCE-SD1-10x5|ReSched-REINFORCE-SD1-20x10|ReSched-PPO-SD1-10x5|ReSched-PPO-SD1-20x10|
> |-|-|-|-|-|-|-|
> |SD1-30x10|5.10|**2.50**|9.26|4.43|3.21|4.50|
> |SD1-40x10|3.65|**1.52**|8.04|3.37|2.19|3.23|
> |SD2-30x10|14.85|11.95|37.76|**8.17**|8.97|11.96|
> |SD2-40x10|0.52|-1.67|23.25|-3.39|**-3.51**|-1.29|
> |Avg.|6.03|3.58|19.58|3.15|**2.72**|4.60|
>
>
> On the open benchmark sets, all DRL-based methods achieve very similar average performance under the small training budget. In particular, the REINFORCE and PPO versions of ReSched trained on small synthetic instances remain competitive with the best DANIEL configuration. This indicates that, with only a small amount of training data and a short training time (e.g., around 20 minutes for PPO version ReSched on SD1-10×5 with our hardware), modern DRL-based methods already reach a reasonably strong level on real-world scheduling benchmarks, highlighting their practical potential for real scheduling applications.
>
> **Table RT8. Performance of DANIEL and ReSched trained with REINFORCE vs. PPO on open FJSP benchmarks under a small training budget. Values are average optimality gaps (%) w.r.t. the best-known upper bounds.**
> |Open Benchmark|DANIEL-SD1-10x5|DANIEL-SD1-15x10|ReSched-REINFORCE-SD1-10x5|ReSched-REINFORCE-SD1-15x10|ReSched-PPO-SD1-10x5|ReSched-PPO-SD1-15x10|
> |-|-|-|-|-|-|-|
> |Brandimarte|13.58|12.97|13.50|14.33|**10.52**|10.77|
> |Hurink(edata)|16.33|**14.41**|18.06|16.34|18.29|20.00|
> |Hurink(rdata)|11.42|12.07|10.28|**9.92**|10.45|13.44|
> |Hurink(vdata)|3.28|3.75|**2.67**|2.82|4.26|3.60|
> |Avg.|11.15|**10.80**|11.13|10.86|10.88|11.95|

---

> ### Author Response · Authors · 2025-12-01
> **ReSched with PPO (part3/3)**
>
> (2). **Large-budget setting**
> In the large-budget setting, we increase the training budget of DANIEL by a factor of 25 under its original configuration, and apply the same 25x scaling to the PPO-based ReSched, so that both methods use roughly 2,000,000 effective trajectory updates, matching the budget of our original REINFORCE experiments.
>
> Under the large-budget setting, both ReSched versions benefit from the increased data, but the PPO version consistently improves over the REINFORCE version across almost all datasets. DANIEL also benefits from the larger budget, yet even the REINFORCE version of ReSched now clearly outperforms DANIEL on most SD2 and medium-sized SD1 instances, and the PPO version achieves the best average in-distribution performance overall. This shows that, even when DANIEL is given a comparable large training budget, the proposed architecture (especially with PPO) remains substantially stronger.
>
> **Table RT9. In-distribution performance of DANIEL and ReSched trained with REINFORCE vs. PPO under a large training budget. Values are average optimality gaps (%) w.r.t. the best-known upper bounds.**
> |In-distribution|DANIEL|ReSched-REINFORCE|ReSched-PPO|
> |-|-|-|-|
> |SD1-10x05|**9.22**|12.25|11.48|
> |SD1-20x05|**3.08**|4.63|4.20|
> |SD1-15x10|10.84|6.51|**5.21**|
> |SD1-20x10|**-0.43**|0.48|-0.36|
> |SD2-10x05|24.75|16.36|**14.24**|
> |SD2-20x05|8.86|9.87|**6.83**|
> |SD2-15x10|53.94|18.14|**16.73**|
> |SD2-20x10|28.89|14.18|**13.79**|
> |Avg.|17.39|10.30|**9.02**|
>
> In the out-of-distribution setting with a large training budget, both ReSched versions clearly outperform DANIEL on average, and the PPO version consistently improves over the REINFORCE version across most datasets.
>
> **Table RT10. Out-of-distribution generalization of DANIEL and ReSched trained with REINFORCE vs. PPO under a large training budget. Values are average optimality gaps (%) w.r.t. the best-known upper bounds.**
> |Out-of-distribution|DANIEL-SD1-10x5|DANIEL-SD1-20x10|ReSched-REINFORCE-SD1-10x5|ReSched-REINFORCE-SD1-20x10|ReSched-PPO-SD1-10x5|ReSched-PPO-SD1-20x10|
> |-|-|-|-|-|-|-|
> |SD1-30x10|2.05|**1.45**|3.44|2.69|2.81|1.91|
> |SD1-40x10|0.98|**0.53**|2.54|1.64|2.06|1.45|
> |SD2-30x10|21.51|18.59|8.79|6.30|**5.16**|7.05|
> |SD2-40x10|4.60|0.05|-2.40|-4.58|**-6.02**|**-6.02**|
> |Avg.|7.29|5.16|3.09|1.51|**1.00**|1.10|
>
>
> Interestingly, when we increase the training budget of DANIEL by 25×, its OOD performance on the SD2 datasets does not improve and even degrades compared to the original setting. This suggests that, under the current training setup, DANIEL does not clearly benefit from additional data, which may partly explain why the original work chose a relatively small training budget. In contrast, ReSched continues to improve when the training budget is increased, indicating that our architecture can effectively leverage more trajectories.
>
> **Table RT11. Out-of-distribution performance of DANIEL under original and large training budgets. Values are average optimality gaps (%) w.r.t. the best-known upper bounds.**
> |Out-of-distribution|DANIEL-SD1-10x5-Origin|DANIEL-SD1-20x10-Origin|DANIEL-SD1-10x5-25000|DANIEL-SD1-20x10-25000|
> |-|-|-|-|-|
> |SD2-30x10|14.85|11.95|21.51|18.59|
> |SD2-40x10|0.52|-1.67|4.60|0.05|
>
> On the open benchmark, all DRL-based methods achieve very similar performance under the large training budget, with only minor differences in gaps. ReSched (both REINFORCE and PPO) remains competitive with DANIEL, indicating that models trained on synthetic SD1 instances transfer reasonably well to these open benchmark and do not exhibit noticeable overfitting to the synthetic distribution.
>
> **Table RT12. Performance of DANIEL and ReSched trained with REINFORCE vs. PPO on open FJSP benchmarks under a large training budget. Numbers are average optimality gaps (%) w.r.t. the best-known upper bounds.**
> |Open Benchmark|DANIEL-SD1-10x5|DANIEL-SD1-20x10|ReSched-REINFORCE-SD1-10x5|ReSched-REINFORCE-SD1-15x10|ReSched-PPO-SD1-10x5|ReSched-PPO-SD1-15x10|
> |-|-|-|-|-|-|-|
> |Brandimarte|14.10|14.58|**9.08**|12.49|10.34|10.97|
> |Hurink(edata)|**14.67**|15.71|15.48|16.34|16.25|18.16|
> |Hurink(rdata)|11.20|10.34|**10.18**|10.31|10.59|10.42|
> |Hurink(vdata)|3.17|3.42|3.48|**2.55**|6.41|3.98|
> |Avg.|10.79|11.01|**9.56**|10.42|10.90|10.88|

---

> ### Author Response · Authors · 2025-12-01
> **Contribution of ReSched**
>
> > **W3**: The contribution of ReSched is marginal in my opinion, since it only introduces a new state representation and network architecture. Moreover, due to previously stated weaknesses, I am not fully convinced about the results and, consequently, the significance of these novel parts.
>
> Our goal is to take a step towards a more general “foundation-style” model for scheduling, where a single framework can handle different scheduling variants without problem-specific engineering. From this perspective, state representation and network architecture are exactly the key components.
>
> (1). Unified view of classic scheduling problems
> We first start from three classic scheduling problems: JSSP, FJSP, and FFSP. We propose that, from a heterogeneous graph perspective (see Fig. 4 in the appendix), JSSP and FFSP can be regarded as special cases of FJSP. This unified perspective motivates us to take FJSP as a generic formulation and to design ReSched on top of this common representation.
>
> (2). Minimal, problem-agnostic state representation:
> To further eliminate the gap between different scheduling problem. We first observe that recent DRL-based methods for FJSP rely on increasingly complex hand-crafted features, from 10 in HGNN to 26 in DANIEL. To check whether the performance of these methods mainly comes from such rich features, we conduct an ablation study on DANIEL (see in Appendix B.3), that a large number of hand-craft features are not a necessary condition for solving FJSP.
>
> Next, we revisit the scheduling formulation to clarify which information is truly essential for the state representation. Through this analysis, we identify four types of information that are sufficient for decision making: available time, duration information, and two graph structures that encode operation precedence and operation–machine feasibility, respectively.
>
> Finally, from a construction perspective, we introduce the subproblem and consistently implemented this formulation in the design of the state part. Concretely, we use relative available times and backward-looking O2O connections to eliminate past decisions while preserving the Markov property. This yields a minimal yet Markov-sufficient state representation that can be shared across different scheduling problems.
>
> (3). Structure-aware network design
> In the network design, we deliberately adopt a Transformer-based architecture instead of the more commonly used GNN-based models. Transformers scale well with larger models and training budgets on modern accelerators[AR15],  which makes our framework well prepared for scaling to larger instances and additional scheduling variants.
>
> However, directly applying Transformer to scheduling is not sufficient. We introduce several structural adaptations that are better aligned with scheduling context, such as dual operation/machine branches design. At the same time, we follow the same minimal-design principle as in our state representation, for example using RoPE as a lightweight positional encoding rather than problem-specific features. When designing these components, we deliberately avoid problem-specific heuristics, so that the resulting framework can be quickly adapted to different scheduling variants.
>
> Finally, the additional multi-seed and training-budget experiments discussed above are intended to directly address the reviewer’s concerns about the robustness and fairness of our results, and they consistently support the conclusions drawn in the paper.
>
> [AR15]. Kaplan, Jared, et al. Scaling laws for neural language models. arXiv,2020.

---

> ### Author Response · Authors · 2025-12-01
> **Discount factor and proposition**
>
> > **Q1**: Could the authors explain why $\gamma = 0.99$, when it is normally set to $\gamma = 0.99$, such as for L2D and DANIEL?
>
> In our case, we initially set the discount factor to 0.99 following a standard deep RL setting, and we found empirically that ReSched is not sensitive to this hyperparameter, so we kept this value in our main configuration. To further address the reviewer’s concern, we reran three additional experiments with $\gamma$ = 1 and compared them with our previous multi-seed results under $\gamma$ = 0.99; the results are very similar across all runs, confirming that our method is indeed insensitive to this hyperparameter.
>
> **Table RT17. Effect of the discount factor $\gamma$ on the performance of ReSched. Values are average optimality gaps (%) w.r.t. the best-known upper bounds. “ReSched-0.99-Avg.” and “ReSched-1.0-Avg.” denote averages over four and three seeds, respectively.**
> |In-distribution|ReSched-0.99-Avg.|ReSched-1.0-Avg.|ReSched-1.0-seed1|ReSched-1.0-seed2|ReSched-1.0-seed3|
> |-|-|-|-|-|-|
> |SD1-10x05|**11.75**|**11.75**|11.41|11.76|12.07|
> |**Out-of-distribution**|**ReSched-0.99-Avg.**|**ReSched-1.0-Avg.**|**ReSched-1.0-seed1**|**ReSched-1.0-seed2**|**ReSched-1.0-seed3**|
> |SD1-30x10|**3.10**|**3.10**|2.63|3.69|2.97|
> |SD1-40x10|2.23|**2.19**|2.00|2.51|2.06|
> |SD2-30x10|**9.04**|10.03|8.28|11.58|10.22|
> |SD2-40x10|**-2.61**|-2.09|-3.55|-0.11|-2.61|
> |Avg.|**2.94**|3.31|2.34|4.42|3.16|
> |**Open Benchmark**|**ReSched-0.99-Avg.**|**ReSched-1.0-Avg.**|**ReSched-1.0-seed1**|**ReSched-1.0-seed2**|**ReSched-1.0-seed3**|
> |Brandimarte|**10.28**|11.52|12.44|10.32|11.80|
> |Hurink(edata)|**15.84**|16.26|16.03|16.89|15.85|
> |Hurink(rdata)|**10.13**|10.50|10.49|10.11|10.91|
> |Hurink(vdata)|2.90|**2.67**|2.66|2.37|2.98|
> |Avg.|**9.78**|10.24|10.41|9.92|10.39|
>
> > **Q4**:In Table 3, it is unclear whether ReSched uses the greedy or sampling evaluation for JSP. Could you please clarify which one is used?
>
> We apologize for the lack of clarity. In Table 3, the results reported for ReSched on the JSSP benchmarks are obtained using the greedy decoding strategy. We have updated the Section 5 of the revised manuscript to explicitly state that JSSP results are evaluated with the greedy strategy.

---

### Official Review · Reviewer_54We · 2025-10-29

**Soundness:** 3
**Presentation:** 3
**Contribution:** 2
**Rating:** 6
**Confidence:** 4

**Summary:**

This paper presents a new neural network architecture and training procedure for different types of scheduling problems, with a particular focus on the FJSSP. The authors also discuss applications to the classical JSSP and the Flow Shop Scheduling Problem.

The main idea of this work is to employ a simplified feature space as input encoding for a transformer-based architecture that uses a non-learnable positional encoding. In contrast to previous research that relies on extensive and often redundant feature representations, the authors deliberately restrict the input to a small set of features that effectively capture the current system state without incorporating information about past actions. This enables the network to focus on a specific sub-problem at each action step.

Through extensive experiments, the authors demonstrate that their method outperforms several existing approaches and is capable of establishing new benchmark results across a variety of problem instances and sizes. The model is trained using deep reinforcement learning with the REINFORCE algorithm, where advantage estimates are incorporated into the policy gradient computation.

The authors also state that they plan to release their implementation publicly.

**Strengths:**

- The paper strikes a balanced narrative between reviewing foundational concepts of the FJSSP and introducing the method. Explanations are concise and accessible, enabling readers to follow both theoretical background and model design decisions.

- Although the transformer architecture itself is not new, the authors extend it with relevant innovations (such as rotary positional encodings) and tailor the input representation to the scheduling context. Further, the authors introduces a deliberate feature reduction strategy, focusing only on state-relevant features, a nontrivial design choice in scheduling, where feature selection is often a bottleneck. In sum, the application and evaluation of these architectural modifications in FJSSP (and its extensions) represent a creative adaptation of transformer models to a new domain.

- The paper provides a solid theoretical justification for using a reduced feature space as input to the transformer network. The claims are supported by a direct comparison with another SotA method that uses a richer feature representation, showing improved performance under feature restriction. Comprehensive ablation studies isolate and quantify the contribution of architectural components.

- The proposed method consistently outperforms strong baselines in extensive experiments including both greedy and sampling-based inference strategies. The authors demonstrate a generalization to classical JSSP and FSSP indicating high potential for broader applicability beyond the initial problem formulation. The results contribute actionable insights to the research community, emphasizing the importance of carefully chosen feature representations in neural approaches to scheduling.

**Weaknesses:**

- The experimental evaluation lacks a comparison of computation time between the proposed method and other neural network–based approaches. This is particularly relevant because the method supports sampling-based inference, which may introduce higher computational cost. A runtime analysis (such as wall-clock time per episode or inference step) would clarify the trade-off between solution quality and computational efficiency and help position the method against deep learning baselines.

- In Figure 1a, the use of plus signs suggests that both solutions are combined to generate a schedule, although the solutions seem independent and not actually aggregated. This may mislead readers. In Figure 2, certain elements, especially the graphs in the upper part of the decision-making module, are not clearly explained in the text, making their meaning and role ambiguous. Enhancing these figures or adding brief explanations in the caption would improve interpretability.

- In the appendix, the authors show that the DANIEL network improves notably when fewer features are used. Table 1 further indicates that DANIEL is generally the second-best method and even outperforms the proposed approach in certain cases. However, the paper does not evaluate how a feature-reduced DANIEL network would perform directly against the proposed model. Since the central contribution of the paper is the effectiveness of a reduced feature space, a head-to-head comparison with a feature-restricted DANIEL baseline would be essential to validate that the performance gain is due to the proposed method, and not merely an effect of feature reduction.

- Appendix C presents a key–value cache mechanism, but its relevance to the main paper is unclear, as no inference-time results are reported. Without linking the cache mechanism to performance measurements, the section feels isolated rather than integrated into the contribution. Including inference-time evaluation, or explicitly discussing when and how the cache is beneficial, would strengthen the connection to the main content.

**Questions:**

1. In Appendix B.3, could you clarify whether the machine embeddings are entirely removed, or whether only the global descriptors are excluded? If no machine features are present, it is unclear how the Dual Attention Network would function. Were these embeddings perhaps replaced with placeholder values, such as zero entries?
2. How does your method compare to the DANIEL network when both use a reduced feature space? In the appendix, you demonstrate that the DANIEL network improves in performance when fewer features are used. Furthermore, Table 1 shows that the DANIEL network generally performs as the second-best method and even outperforms your proposed approach in certain cases. How would a feature-reduced version of the DANIEL network compare directly to your method?
3. The paper presents several ablation studies analyzing various components of the proposed model, but the choice of the REINFORCE algorithm appears to have been assumed without direct justification. Since all competing methods were trained using a Proximal Policy Optimization algorithm, was this alternative also tested? If so, did it yield inferior results? Including such a comparison in the ablation studies would strengthen the empirical analysis.
4. How does the formulation of the reward function impact the training process? This question is particularly relevant since the DANIEL method learns based on the change in the current makespan rather than the change in the expected lower bound.
5. Why was the inference speed presented in only a single figure and not compared across all experiments?
6. Why were no ablation studies conducted on the network size, such as the number of attention heads or Transformer block dimensions? This seems highly relevant when introducing a new architecture.
7. Why was the RoPE positional encoding chosen? Were other positional encoding methods considered, and how might their use impact learning performance, particularly regarding inference speed and makespan?
8. A variety of metaheuristics exist for solving the FJSP. Why were none of these methods investigated or compared against the proposed approach?
9. The FJSP, JSSP, and FFSP represent relatively limited and less complex scheduling problems. How could this approach be extended to handle a broader range of scheduling problems, including those with more diverse and realistic constraints encountered in real-world applications?

---

> ### Comment · Reviewer_54We · 2025-11-27
>
> Since the authors did not submit a rebuttal or clarifications, I maintain my original assessment. This is unfortunate, as several of the raised questions by all reviewers could have benefited from further explanation. A response might have helped to strengthen the contribution and address remaining uncertainties.

---

> > ### Author Response · Authors · 2025-11-27
> > **Thank you for your attention on our response**
> >
> > Dear Reviewer 54We,
> >
> > Thank you for your ongoing attention and guidance on our paper. We are currently finalizing our responses to your comments as well as those from the other reviewers. We sincerely apologize for the delay and are committed to addressing all feedback thoroughly. We will post our response very soon.
> >
> > Sincerely,
> > The Authors

---

> ### Author Response · Authors · 2025-12-01
>
> > **W1**: The experimental evaluation lacks a comparison of computation time between the proposed method and other neural network–based approaches. This is particularly relevant because the method supports sampling-based inference, which may introduce higher computational cost. A runtime analysis (such as wall-clock time per episode or inference step) would clarify the trade-off between solution quality and computational efficiency and help position the method against deep learning baselines.
>
> > **Q5**: Why was the inference speed presented in only a single figure and not compared across all experiments?
>
> Thanks for the reviewer's concern about runtime analysis. In the main paper, we chose not to report wall-clock time in every result table, because adding runtime for all methods and all settings in Table 1 (both greedy and sampling) would nearly double its size and leave much less space for reporting solution-quality results. Instead, we focused the main tables on solution quality and used a separate figure for runtime.
>
> For inference speed, we selected the open FJSP benchmarks (Brandimarte and Hurink), which cover a wide range of instance sizes (jobs from 10 to 30, machines  from 4 to 15). Figure 3 in the paper summarizes the average inference time of all methods on all open benchmarks under this setting. The results show that ReSched has a **comparable** runtime to the DRL baselines (HGNN and DANIEL), while achieving noticeably better solution quality.
>
> Regarding the sampling strategy, we follow the same setting as prior DRL methods (HGNN and DANIEL): for each test instance, the stochastic policy is run multiple times with independent samples and the best solution is reported. We have clarified this procedure in Sec. 5 of the revised manuscript.
>
> To further address the reviewer’s concern about the computational cost of sampling-based inference, we additionally report the average wall-clock inference time (in seconds) for the two DRL baselines and ReSched on all open benchmarks, measured on the same hardware as in the main experiments:
>
> **Table RT5. Sampling strategy inference runtime of ReSched on FJSP open benchmarks.**
> |runtime(s)|HGNN|DANIEL|ReSched|
> |-|-|-|-|
> |Brandimarte|3.31|3.15|3.17|
> |Hurink(edata)|3.39|3.41|3.51|
> |Hurink(rdata)|3.77|3.63|3.43|
> |Hurink(vdata)|3.82|3.72|3.69|
>
> > **W2**: In Figure 1a, the use of plus signs suggests that both solutions are combined to generate a schedule, although the solutions seem independent and not actually aggregated. This may mislead readers. In Figure 2, certain elements, especially the graphs in the upper part of the decision-making module, are not clearly explained in the text, making their meaning and role ambiguous. Enhancing these figures or adding brief explanations in the caption would improve interpretability.
>
> We appreciate the reviewer’s suggestion regarding the clarity of Figures 1 and 2. In the revised manuscript, we have modified Fig. 1(a) to remove the misleading “+” symbol and now explicitly illustrate that the sampled solutions are evaluated independently rather than aggregated. For Fig. 2, we have updated the caption to more clearly explain the decision-making module: how the operation, machine and edge embeddings for each feasible operation–machine pair are concatenated, scored by an MLP. We hope these changes make the figures easier to interpret.
>
> > **Q1**: In Appendix B.3, could you clarify whether the machine embeddings are entirely removed, or whether only the global descriptors are excluded? If no machine features are present, it is unclear how the Dual Attention Network would function. Were these embeddings perhaps replaced with placeholder values, such as zero entries?
>
> Thanks for pointing out this ambiguity. The ablation in Appendix B.3 is performed step by step, and the two rows in Table 4 correspond to different levels of removal.
>
> In the first row (“Del Global”), we only remove the global descriptors from the actor network, while keeping the machine branch and all machine-node embeddings exactly the same as in the original DANIEL implementation.
>
> In the second row (“Del Global Machine”), we remove the global embedding and machine embeddings simultaneously from the actor, i.e., we disable the machine branch and compute the action scores using only the operation branch with the O2M mask.
>
> In this configuration the policy network no longer uses any machine features; the dual GAT-style architecture effectively degenerates into a single operation branch. We do not replace the machine embeddings with placeholder or zero vectors, the outputs of the machine branch are simply not used when computing the policy logits.
>
> This design shows that, on SD2-15×10, even when the entire machine branch of DANIEL is removed, the in-distribution performance remains almost unchanged, indicating that nearly half of the original network (the machine branch) does not contribute to the final solution quality under this setting.

---

> ### Author Response · Authors · 2025-12-01
>
> > **W3**: In the appendix, the authors show that the DANIEL network improves notably when fewer features are used. Table 1 further indicates that DANIEL is generally the second-best method and even outperforms the proposed approach in certain cases. However, the paper does not evaluate how a feature-reduced DANIEL network would perform directly against the proposed model. Since the central contribution of the paper is the effectiveness of a reduced feature space, a head-to-head comparison with a feature-restricted DANIEL baseline would be essential to validate that the performance gain is due to the proposed method, and not merely an effect of feature reduction.
>
> > **Q2**: How does your method compare to the DANIEL network when both use a reduced feature space? In the appendix, you demonstrate that the DANIEL network improves in performance when fewer features are used. Furthermore, Table 1 shows that the DANIEL network generally performs as the second-best method and even outperforms your proposed approach in certain cases. How would a feature-reduced version of the DANIEL network compare directly to your method?
>
> We appreciate the reviewer’s suggestion to compare our method against a feature–reduced（with only 5 features） version of DANIEL. To address this, we followed the ablation setting on DANIEL from Appendix B.3: we trained the original DANIEL and its feature-reduced variant (denoted as DANIEL-Del) on SD2-15×10. To comprehensively evaluate their performance and generalization, we test the models trained on SD2-15×10 (DANIEL, DANIEL-Del, and ReSched) on in-distribution, out-of-distribution synthetic instances, and the open FJSP benchmarks; the detailed results are reported in Table RT6.
>
> **Table RT5. Comparison of ReSched with DANIEL and its feature-reduced variant(DANIEL-Del). Values are average optimality gaps (%) w.r.t. the best-known upper bounds.**
> |In-distribution|DANIEL|DANIEL-Del|ReSched|
> |-|-|-|-|
> |SD2-15x10|56.28|55.81|**18.14**|
> |**Out-of-distribution**|**DANIEL**|**DANIEL-Del**|**ReSched**|
> |SD1-30x10|15.23|**3.84**|5.68|
> |SD1-40x10|**2.73**|3.96|4.18|
> |SD2-30x10|14.58|13.54|**6.62**|
> |SD2-40x10|-0.03|-2.06|**-3.20**|
> |Avg.|8.13|4.82|**3.32**|
> |**Open Benchmark**|**DANIEL**|**DANIEL-Del**|**ReSched**|
> |Brandimarte|17.15|17.54|**13.02**|
> |Hurink(edata)|19.52|24.85|**15.41**|
> |Hurink(rdata)|21.56|22.15|**11.30**|
> |Hurink(vdata)|14.92|14.42|**4.56**|
> |Avg.|18.29|19.74|**11.07**|
>
> On the in-distribution SD2-15×10 setting, DANIEL-Del achieves almost the same performance as the original DANIEL, while ReSched yields a much smaller optimality gap. This indicates that simply reducing the feature set does not close the gap between DANIEL and our method. On the out-of-distribution synthetic instances, DANIEL-Del does improve over DANIEL on average (indicating that overly complex features may hurt generalization), but ReSched still achieves the lowest average gap over all out-of-distribution sizes. Finally, on the open benchmarks, ReSched consistently outperforms both DANIEL and DANIEL-Del.
>
> Overall, these results confirm that while feature reduction can modestly improve DANIEL in some cases, it is not sufficient to match the performance of ReSched. The main gains come from the combination of our minimal yet sufficient state representation and the proposed Transformer-based architecture, rather than from feature reduction alone.
>
> > **W4**: Appendix C presents a key–value cache mechanism, but its relevance to the main paper is unclear, as no inference-time results are reported. Without linking the cache mechanism to performance measurements, the section feels isolated rather than integrated into the contribution. Including inference-time evaluation, or explicitly discussing when and how the cache is beneficial, would strengthen the connection to the main content.
>
> We appreciate the reviewer’s recognition of the potential of our key–value cache mechanism in the ReSched framework. To further clarify its potential, we have included a computational complexity analysis in the revised manuscript (see Appendix. C), which demonstrates that, under common scheduling settings, the use of the KV cache could reduce the computational cost of attention by more than an order of magnitude.
>
> That said, integrating and thoroughly benchmarking this optimization across all experimental settings would require non-trivial engineering effort and additional compute, and we were not able to complete this within the rebuttal period. To avoid overstating its role, we do not intend to include it as a central contribution of the paper but rather as a promising avenue for ReSched.

---

> ### Author Response · Authors · 2025-12-01
> **ReSched PPO version(part1/3)**
>
> > **Q3**: The paper presents several ablation studies analyzing various components of the proposed model, but the choice of the REINFORCE algorithm appears to have been assumed without direct justification. Since all competing methods were trained using a Proximal Policy Optimization algorithm, was this alternative also tested? If so, did it yield inferior results? Including such a comparison in the ablation studies would strengthen the empirical analysis.
>
> We thank the reviewer for raising this concern. Our main contribution lies in the state representation and network architecture rather than in the choice of a particular RL algorithm. In the initial version, we adopted a REINFORCE-style policy-gradient method mainly for its simplicity and because it follows standard practice in Transformer-based neural combinatorial optimization, such as AM[AR10] and POMO[AR11], where vanilla REINFORCE is commonly used to train the policy network. In the revised manuscript (Sec. 4.2.2) we now discuss this design choice and clarify our motivation for using REINFORCE in the initial version.
>
> To further evaluate the effectiveness of our method with different RL algorithms, we implemented a PPO version of ReSched. For a fair comparison, we conducted additional experiments under two training settings: a **small-budget** setting matched to DANIEL, and a **large-budget setting** comparable to our original configuration. For a comprehensive assessment, we follow the same evaluation protocol as in the main paper and evaluate under a greedy strategy on in-distribution, out-of-distribution, and open benchmark.
>
> These new experiments show that our architecture works well with both REINFORCE and PPO: the PPO version converges faster and further improves average in-distribution and out-of-distribution performance, while the REINFORCE version already remains competitive or better than DANIEL under matched budgets. Overall, across all budgets and RL algorithms, ReSched consistently matches or outperforms DANIEL, indicating that the benefits come from our framework rather than from the large amount of training data.

---

> ### Author Response · Authors · 2025-12-01
> **ReSched PPO version(part2/3)**
>
> (1). **Small-budget setting**
> In the small-budget setting, we retrain the REINFORCE-based ReSched with all other hyperparameters unchanged, while reducing the number of epochs so that the total number of trajectories is 80 × 1000 = 80,000, which is on the same order as DANIEL. For the PPO version of ReSched, we use 400 epochs with a batch size of 50, and reuse each collected batch for $K$ = 4 PPO updates (the same $K$ as in DANIEL), again resulting in 400 × 50 × 4 = 80,000 effective trajectory updates. The PPO version of ReSched adopts exactly the same critic architecture and PPO hyperparameters as DANIEL.
>
> Under this small training budget, the PPO version converges faster and achieves better in-distribution performance than both DANIEL and the REINFORCE version of ReSched. Even the REINFORCE version, despite its slower convergence, still surpasses DANIEL in terms of average optimality gap, indicating that the main gain comes from our architecture rather than from using a larger training budget. PPO further exploits the limited data more efficiently than REINFORCE, yielding the best average results among all compared methods.
>
> **Table RT6. In-distribution performance of DANIEL and ReSched trained with REINFORCE vs. PPO under a small training budget. Values are average optimality gaps (%) w.r.t. the best-known upper bounds.**
> |In-distribution|DANIEL|ReSched-REINFORCE|ReSched-PPO|
> |-|-|-|-|
> |SD1-10x05|**10.87**|14.61|11.20|
> |SD1-20x05|**5.03**|8.51|5.84|
> |SD1-15x10|12.42|12.91|**9.41**|
> |SD1-20x10|**1.31**|6.97|3.50|
> |SD2-10x05|25.68|19.06|**15.77**|
> |SD2-20x05|11.52|10.76|**9.21**|
> |SD2-15x10|57.16|**24.03**|25.015|
> |SD2-20x10|31.58|**17.913**|19.38|
> |Avg.|19.45|14.35|**12.42**|
>
> In the out-of-distribution setting, models trained on SD1-10×5 and SD1-20×10 are evaluated on larger unseen instances. Under the small training budget, the PPO version trained only on SD1-10×5 already achieves the best average OOD performance, showing that our architecture combined with PPO can generalize well even when trained on the smallest problem size with limited data. In contrast, the REINFORCE version trained on SD1-10×5 generalizes poorly under this small budget, which is attributed to its lower sample efficiency. However, when the training size is increased to SD1-20×10 (with the same budget), its OOD performance improves substantially and becomes comparable to DANIEL. Overall, these results indicate that our architecture does generalize beyond the training size, with PPO exploiting limited data more efficiently (as PPO trains the policy multiple times on the same data), whereas REINFORCE requires somewhat richer training instances to reach a similar level of OOD performance.
>
> **Table RT7. Out-of-distribution generalization of DANIEL and ReSched trained with REINFORCE vs. PPO under a small training budget. Values are average optimality gaps (%) w.r.t. the best-known upper bounds.**
> |Out-of-distribution|DANIEL-SD1-10x5|DANIEL-SD1-20x10|ReSched-REINFORCE-SD1-10x5|ReSched-REINFORCE-SD1-20x10|ReSched-PPO-SD1-10x5|ReSched-PPO-SD1-20x10|
> |-|-|-|-|-|-|-|
> |SD1-30x10|5.10|**2.50**|9.26|4.43|3.21|4.50|
> |SD1-40x10|3.65|**1.52**|8.04|3.37|2.19|3.23|
> |SD2-30x10|14.85|11.95|37.76|**8.17**|8.97|11.96|
> |SD2-40x10|0.52|-1.67|23.25|-3.39|**-3.51**|-1.29|
> |Avg.|6.03|3.58|19.58|3.15|**2.72**|4.60|
>
>
> On the open benchmark sets, all DRL-based methods achieve very similar average performance under the small training budget. In particular, the REINFORCE and PPO versions of ReSched trained on small synthetic instances remain competitive with the best DANIEL configuration. This indicates that, with only a small amount of training data and a short training time (e.g., around 20 minutes for PPO version of ReSched on SD1-10×5 with our hardware), modern DRL-based methods already reach a reasonably strong level on real-world scheduling benchmarks, highlighting their practical potential for real scheduling applications.
>
> **Table RT8. Performance of DANIEL and ReSched trained with REINFORCE vs. PPO on open FJSP benchmarks under a small training budget. Values are average optimality gaps (%) w.r.t. the best-known upper bounds.**
> |Open Benchmark|DANIEL-SD1-10x5|DANIEL-SD1-15x10|ReSched-REINFORCE-SD1-10x5|ReSched-REINFORCE-SD1-15x10|ReSched-PPO-SD1-10x5|ReSched-PPO-SD1-15x10|
> |-|-|-|-|-|-|-|
> |Brandimarte|13.58|12.97|13.50|14.33|**10.52**|10.77|
> |Hurink(edata)|16.33|**14.41**|18.06|16.34|18.29|20.00|
> |Hurink(rdata)|11.42|12.07|10.28|**9.92**|10.45|13.44|
> |Hurink(vdata)|3.28|3.75|**2.67**|2.82|4.26|3.60|
> |Avg.|11.15|**10.80**|11.13|10.86|10.88|11.95|

---

> ### Author Response · Authors · 2025-12-01
> **ReSched PPO version(part3/3)**
>
> (2). **Large-budget setting**
> In the large-budget setting, we increase the training budget of DANIEL by a factor of 25 relative to its original configuration, and apply the same 25x scaling to the PPO-based ReSched, so that both methods use roughly 2,000,000 effective trajectory updates, matching the budget of our original REINFORCE experiments.
>
> Under the large-budget setting, both ReSched versions benefit from the increased data, but the PPO version consistently improves over the REINFORCE version across almost all datasets. DANIEL also benefits from the larger budget, yet even the REINFORCE version of ReSched now clearly outperforms DANIEL on most SD2 and medium-sized SD1 instances, and the PPO version achieves the best average in-distribution performance overall. This shows that, even when DANIEL is given a comparable large training budget, the proposed architecture (especially with PPO) remains substantially stronger.
>
> **Table RT9. In-distribution performance of DANIEL and ReSched trained with REINFORCE vs. PPO under a large training budget. Values are average optimality gaps (%) w.r.t. the best-known upper bounds.**
> |In-distribution|DANIEL|ReSched-REINFORCE|ReSched-PPO|
> |-|-|-|-|
> |SD1-10x05|**9.22**|12.25|11.48|
> |SD1-20x05|**3.08**|4.63|4.20|
> |SD1-15x10|10.84|6.51|**5.21**|
> |SD1-20x10|**-0.43**|0.48|-0.36|
> |SD2-10x05|24.75|16.36|**14.24**|
> |SD2-20x05|8.86|9.87|**6.83**|
> |SD2-15x10|53.94|18.14|**16.73**|
> |SD2-20x10|28.89|14.18|**13.79**|
> |Avg.|17.39|10.30|**9.02**|
>
> In the out-of-distribution setting with a large training budget, both ReSched versions clearly outperform DANIEL on average, and the PPO version consistently improves over the REINFORCE version across most datasets.
>
> **Table RT10. Out-of-distribution generalization of DANIEL and ReSched trained with REINFORCE vs. PPO under a large training budget. Values are average optimality gaps (%) w.r.t. the best-known upper bounds.**
> |Out-of-distribution|DANIEL-SD1-10x5|DANIEL-SD1-20x10|ReSched-REINFORCE-SD1-10x5|ReSched-REINFORCE-SD1-20x10|ReSched-PPO-SD1-10x5|ReSched-PPO-SD1-20x10|
> |-|-|-|-|-|-|-|
> |SD1-30x10|2.05|**1.45**|3.44|2.69|2.81|1.91|
> |SD1-40x10|0.98|**0.53**|2.54|1.64|2.06|1.45|
> |SD2-30x10|21.51|18.59|8.79|6.30|**5.16**|7.05|
> |SD2-40x10|4.60|0.05|-2.40|-4.58|**-6.02**|**-6.02**|
> |Avg.|7.29|5.16|3.09|1.51|**1.00**|1.10|
>
> Interestingly, when we increase the training budget of DANIEL by 25×, its OOD performance on the SD2 datasets does not improve and even degrades compared to the original setting. This suggests that, under the current training setup, DANIEL does not clearly benefit from additional data, which may partly explain why the original work chose a relatively small training budget. In contrast, ReSched continues to improve when the training budget is increased, indicating that our architecture can effectively leverage more trajectories.
>
> **Table RT11. Out-of-distribution performance of DANIEL under original and large training budgets. Values are average optimality gaps (%) w.r.t. the best-known upper bounds.**
> |Out-of-distribution|DANIEL-SD1-10x5-Origin|DANIEL-SD1-20x10-Origin|DANIEL-SD1-10x5-25000|DANIEL-SD1-20x10-25000|
> |-|-|-|-|-|
> |SD2-30x10|14.85|11.95|21.51|18.59|
> |SD2-40x10|0.52|-1.67|4.60|0.05|
>
> On the open benchmark, all DRL-based methods achieve very similar performance under the large training budget, with only minor differences in gaps. ReSched (both REINFORCE and PPO) remains competitive with DANIEL, indicating that models trained on synthetic SD1 instances transfer reasonably well to these open benchmark and do not exhibit noticeable overfitting to the synthetic distribution.
>
> **Table RT12. Performance of DANIEL and ReSched trained with REINFORCE vs. PPO on open FJSP benchmarks under a large training budget. Numbers are average optimality gaps (%) w.r.t. the best-known upper bounds.**
> |Open Benchmark|DANIEL-SD1-10x5|DANIEL-SD1-20x10|ReSched-REINFORCE-SD1-10x5|ReSched-REINFORCE-SD1-15x10|ReSched-PPO-SD1-10x5|ReSched-PPO-SD1-15x10|
> |-|-|-|-|-|-|-|
> |Brandimarte|14.10|14.58|**9.08**|12.49|10.34|10.97|
> |Hurink(edata)|**14.67**|15.71|15.48|16.34|16.25|18.16|
> |Hurink(rdata)|11.20|10.34|**10.18**|10.31|10.59|10.42|
> |Hurink(vdata)|3.17|3.42|3.48|**2.55**|6.41|3.98|
> |Avg.|10.79|11.01|**9.56**|10.42|10.90|10.88|
>
> [AR10]. Kool W, Van Hoof H, Welling M. Attention, learn to solve routing problems!. International Conference on Learning Representations, 2019.
> [AR11]. Kwon Y D, Choo J, Kim B, et al. Pomo: Policy optimization with multiple optima for reinforcement learning. Advances in Neural Information Processing Systems, 2020

---

> ### Author Response · Authors · 2025-12-01
>
> > **Q4**: How does the formulation of the reward function impact the training process? This question is particularly relevant since the DANIEL method learns based on the change in the current makespan rather than the change in the expected lower bound.
>
> We thank the reviewer for raising this important point. We would like to clarify that, contrary to what was stated in the question, the DANIEL method in fact uses the Estimate Lower Bound Reward, rather than the change in the current makespan (delta makespan).
>
> In our early framework explorations, we tested three different reward formulations: delta makespan, estimated mean reward, and estimated lower-bound reward. As summarized in Table RT13, all three variants are able to train a reasonable policy, but the choice of reward has a noticeable impact on generalization. In particular, the delta-makespan reward tends to fit the synthetic distribution more aggressively: it can slightly improve average performance on synthetic OOD instances, but it significantly degrades performance on the open benchmarks compared to the estimated lower-bound reward, which we interpret as a form of overfitting to the synthetic training distribution. As the Estimate Lower Bound Reward is commonly used in DRL based scheduling methods like L2D[AR5] and DANIEL, we also use it to align with the standard practice.
>
> **Table RT13. Effect of different reward on the performance of ReSched. Values are average optimality gaps (%) w.r.t. the best-known upper bounds.**
> |In-distribution|ReSched-Est_LB|ReSched-Est_Mean|ReSched-Delta_Makespan|
> |-|-|-|-|
> |SD1-10x05|**12.25**|12.56|13.05|
> |**Out-of-distribution**|**ReSched-Est_LB**|**ReSched-Est_Mean**|**ReSched-Delta_Makespan**|
> |SD1-30x10|3.44|**3.29**|3.30|
> |SD1-40x10|2.54|**2.15**|2.41|
> |SD2-30x10|8.79|9.19|**7.53**|
> |SD2-40x10|-2.40|-2.57|**-4.30**|
> |Avg.|3.09|3.015|**2.24**|
> |**Open Benchmark**|**ReSched-Est_LB**|**ReSched-Est_Mean**|**ReSched-Delta_Makespan**|
> |Brandimarte|**9.08**|9.66|15.35|
> |Hurink(edata)|**15.48**|16.58|17.27|
> |Hurink(rdata)|**10.18**|10.34|11.62|
> |Hurink(vdata)|3.48|**2.75**|4.5|
> |Avg.|**9.56**|9.83|12.19|
>
> > **Q6**: Why were no ablation studies conducted on the network size, such as the number of attention heads or Transformer block dimensions? This seems highly relevant when introducing a new architecture.
>
> Thank you for raising this point. Regarding the design of the Transformer block hyperparameters, such as the number of attention heads and hidden dimensions, our initial configuration followed the settings from AM [AR10] and POMO [AR11], which are well-established Transformer-based methods for neural combinatorial optimization. These methods have been proven effective in similar tasks, which motivated our choice of hyperparameters.
>
> As for the number of Transformer blocks, we opted for 2 layers to maintain the essential flow of information. This decision was influenced by the use of hop connections in our O2O graph design, which allows the model to effectively capture dependencies across operations with fewer layers. We found this configuration sufficient to model the problem without introducing unnecessary complexity.
>
> [AR5]. Zhang C, Song W, Cao Z, et al. Learning to dispatch for job shop scheduling via deep reinforcement learning. Advances in neural information processing systems, 2020.
>
> [AR10]. Kool W, Van Hoof H, Welling M. Attention, learn to solve routing problems!. International Conference on Learning Representations, 2019.
>
> [AR11]. Kwon Y D, Choo J, Kim B, et al. Pomo: Policy optimization with multiple optima for reinforcement learning. Advances in Neural Information Processing Systems, 2020

---

> ### Author Response · Authors · 2025-12-01
>
> > **Q7**: Why was the RoPE positional encoding chosen? Were other positional encoding methods considered, and how might their use impact learning performance, particularly regarding inference speed and makespan?
>
> Our goal in the network design is to keep the architecture simple and problem-agnostic while still providing the model with basic positional cues. We therefore chose Rotary Positional Embeddings (RoPE) because they are lightweight, parameter-free, and encode relative positions directly in the operation attention scores, which is well suited to variable-size instances and extrapolation to larger problem sizes. In addition, RoPE has been empirically validated in a variety of Transformer applications and has become a widely adopted choice for positional encoding in recent models [AR12].
>
> We also consider and implement an alternative positional encoding in earlier versions of our model, where we treated the operation index as an additional feature and passed it through a learnable linear layer (i.e., a simple learned absolute position feature concatenated to the node embedding). In preliminary experiments on SD1/SD2 training settings, this index-based encoding led to very similar convergence behavior and final makespan as RoPE. As summarized in Table RT14, when training on SD1–10×5, the two variants achieve comparable in-distribution performance, and their average gaps on out-of-distribution synthetic instances and open benchmarks are also close, with RoPE being slightly better overall.
>
> **Table RT14. Effect of positional encoding choice (RoPE vs index-based) on the performance of ReSched. Values are average optimality gaps (%) w.r.t. the best-known upper bounds.**
> |In-distribution|ReSched-RoPE|ReSched-idx|
> |-|-|-|
> |SD1-10x05|12.25|12.74|
> |**Out-of-distribution**|**ReSched-RoPE**|**ReSched-idx**|
> |SD1-30x10|3.44|3.35|
> |SD1-40x10|2.54|2.42|
> |SD2-30x10|8.79|9.95|
> |SD2-40x10|-2.40|-1.88|
> |Avg.|3.01|3.46|
> |**Open Benchmark**|**ReSched-RoPE**|**ReSched-idx**|
> |Brandimarte|9.08|11.05|
> |Hurink(edata)|15.48|15.94|
> |Hurink(rdata)|10.18|11.29|
> |Hurink(vdata)|3.48|2.91|
> |Avg.|9.56|10.29|
>
> In terms of inference speed, RoPE does introduce a modest computational overhead due to the additional rotation in each attention head. We consider this overhead acceptable in view of RoPE’s better extensibility and keep RoPE in the main configuration, as it aligns better with our minimal, problem-agnostic design while maintaining comparable learning performance.
>
> **Table RT15. Inference runtime of ReSched with different positional encodings on open FJSP benchmarks. Values are average inference times per instance in seconds.**
> |Time(s)|with RoPE|without RoPE|with-idx|
> |-|-|-|-|
> |Brandimarte|1.31|1.21|1.19|
> |Hurink(edata)|1.38|1.22|1.24|
> |Hurink(rdata)|1.37|1.22|1.24|
> |Hurink(vdata)|1.37|1.22|1.24|
> |Avg.|1.36|1.22|1.23|
>
> > **Q8**: A variety of metaheuristics exist for solving the FJSP. Why were none of these methods investigated or compared against the proposed approach?
>
> We agree that there is a rich line of metaheuristic research for FJSP, and these methods have contributed many strong best-known solutions. In this work, however, our primary goal is to study a unified DRL-based framework and to compare it against recent learning-based approaches under a common implementation and evaluation protocol. For this reason, the initial version focused on DRL baselines (HGNN, DANIEL, DOAGNN, etc.) together with classical dispatching rules, and reported performance in terms of optimality gaps with respect to the best-known upper bounds (see in Table 1, bottom block with the open-benchmark dataset).
>
> In the revised manuscript, we have additionally included explicit comparisons with representative non-learning methods (see in Table 1). In particular, we add the 2SGA genetic algorithm [AR13], a competitive metaheuristic tailored for FJSP.
>
> While 2SGA achieves very strong solution quality, it typically requires much longer wall-clock time, often in the order of minutes to hours under standard time limits, and relies on problem-specific heuristic design that does not directly transfer to other scheduling variants. By contrast, once trained, ReSched can generate solutions within a few seconds per instance even with a simple sampling strategy (around 3s per instance in our open FJSP benchmarks), and on the more flexible Hurink(vdata) set its solution quality is close to that of 2SGA. This reflects our intended positioning: ReSched is not meant to replace all specialized metaheuristics, but to provide a fast, unified policy that works well across multiple scheduling variants, whereas classical metaheuristics remain powerful but more problem-specific and time-inefficient optimizers.
>
> [AR12]. Su, Jianlin, et al. Roformer: Enhanced transformer with rotary position embedding. Neurocomputing, 2024.
>
> [AR13]. Rooyani, Danial, and Fantahun M. Defersha. An efficient two-stage genetic algorithm for flexible job-shop scheduling. Ifac-Papersonline,2019.

---

> ### Author Response · Authors · 2025-12-01
>
> > **Q9**: The FJSP, JSSP, and FFSP represent relatively limited and less complex scheduling problems. How could this approach be extended to handle a broader range of scheduling problems, including those with more diverse and realistic constraints encountered in real-world applications?
>
> We agree that real-world scheduling often involves richer and more diverse constraints than the standard FJSP/JSSP/FFSP benchmarks. The goal of ReSched is precisely to build a simple and generic framework that can be quickly adapted to different scheduling or resource-allocation scenarios. To this end, we start from FJSP as a generic formulation, view other classic shceudling problems (JSSP, FFSP) through a heterogeneous operation–machine graph, and deliberately avoid problem-specific features or heuristics in both the state representation and the network architecture. This is why the same model can already handle FJSP, JSSP, and FFSP without any architectural changes.
>
> As a concrete example, dependent task offloading in mobile edge computing [AR14] can be viewed as an instance of our formulation. In this problem, an application is modeled as a DAG of computation tasks with precedence constraints, and the scheduler must decide for each ready task whether to execute it locally or offload it to one of several heterogeneous edge/cloud servers, subject to communication and resource limits, in order to minimize end-to-end latency (or a latency–energy trade-off).
>
> In this case，each computation task in the application DAG corresponds to an operation node, and each local device, edge server, or cloud node corresponds to a machine node. Task dependencies form the O2O edges, while the end-to-end latency for executing a task on a given resource (communication plus computation, or purely computation for local execution) defines the O2M edge durations, with infeasible assignments masked out. Under this mapping, the task DAG is directly represented as the O2O connection mask fed into the operation branch, so no architectural change is required. The scheduler’s decision at each step is to select a feasible task–resource pair, and the objective is typically a function of task completion times such as total latency or a latency–energy trade-off. Our state representation and dual-branch Transformer can thus be reused without modification; only the duration model and reward definition need to be adapted, illustrating how ReSched can be extended to more realistic scheduling scenarios.
>
> [AR14]. Wang J, Hu J, Min G, et al. Dependent task offloading for edge computing based on deep reinforcement learning. IEEE Transactions on Computers, 2021.

---

### Official Review · Reviewer_cunA · 2025-10-31

**Soundness:** 3
**Presentation:** 3
**Contribution:** 3
**Rating:** 6
**Confidence:** 4

**Summary:**

The paper claims that a minimal Markov-sufficient state for FJSP can be formed using four core features. It tries to remove historical dependencies, redundancy, and auxiliary variables from node features, and introduces a dual-branch Transformer: an operations branch with self-attention enhanced by RoPE and a machine branch with cross-attention that injects edge features and self-connections to stabilize operation–machine imbalance. The method reports SOTA on FJSP with promising transfer to JSSP/FFSP.

**Strengths:**

The paper shows that strong performance is possible with a small, domain-aware feature set.

Structure-aware Transformer well aligned with precedence and operation-machine eligibility.

It demonstrates sample efficiency and generalization across instance sizes in FJSP, one of the most challenging COPs.

**Weaknesses:**

The “minimal MDP” claim (proposition 1) lacks a formal proof; it relies mainly on empirical results.

Related work on feature minimization is incomplete: Lee & Kim (2024) already aimed to remove historical dependencies and relative time feature (global minimum available time subtraction) in JSSP; conceptually aligned with “State: SubProblem” design (line 229 in this paper). Authors analyze only DANIEL (Wang et al., 2024b) features in FJSP, but feature minimization in JSSP had already been explored.
- Lee, J. H., & Kim, H. J. (2025). Graph-based imitation learning for real-time job shop dispatcher. IEEE T-ASE.

JSP comparisons are incomplete and relatively weak. While FJSP results include several recent methods, JSP needs stronger baselines. For example:
– Park, J., Bakhtiyar, S., & Park, J. (2021). ScheduleNet. arXiv:2106.03051.
– Chen, R., Li, W., & Yang, H. (2022). TII 19(2):1322–1331.
– Iklassov, Z. et al. (2023). IJCAI, pp. 5350–5358.
– Lee, J. H., & Kim, H. J. (2025). Graph-based imitation learning for real-time job shop dispatcher. IEEE T-ASE.

**Questions:**

1. Can you provide a formal proof that the four-feature state is Markov-sufficient (proposition 1)?

2. You excel on FJSP but are less competitive on recent JSP baselines. What structural or distributional differences make JSP harder for your method?

---

> ### Author Response · Authors · 2025-12-01
> **Response (1/3)**
>
> > **W1**: The “minimal MDP” claim (proposition 1) lacks a formal proof; it relies mainly on empirical results.
>
> > **Q1**: Can you provide a formal proof that the four-feature state is Markov-sufficient (proposition 1)?
>
> We thank the reviewer for their insightful comment regarding the "minimal MDP" claim and the need for a formal proof. We have revised the main text to employ more precise terminology, tempering the original informal wording. Furthermore, we have now incorporated a formal proof of Proposition 1 in Appendix D. This proof establishes that our proposed state is Markov-sufficient, demonstrating that the remaining feasible solution set and any objective function dependent solely on operation finish times are determined exclusively by the current state.
>
> > **W2**: Related work on feature minimization is incomplete: Lee & Kim (2024) already aimed to remove historical dependencies and relative time feature (global minimum available time subtraction) in JSSP; conceptually aligned with “State: SubProblem” design (line 229 in this paper). Authors analyze only DANIEL (Wang et al., 2024b) features in FJSP, but feature minimization in JSSP had already been explored. (Lee, J. H., & Kim, H. J. (2025). Graph-based imitation learning for real-time job shop dispatcher. IEEE T-ASE.)
>
> Thanks the reviewer for pointing out these excellent works!  In the revised manuscript, we have added [AR4] to the discussion of state representation in Sec. 4.1.
>
> We agree that simplifying the state representation is an important step towards more general, foundation-style models for scheduling. The work of Lee and Kim[AR4] is an interesting example in this direction for JSSP.
>
> However, their design remains tailored to the JSSP setting.
>
> First, their node representation still consists of eight hand-crafted features. For example, the “machine delay to start” $g_{o_\tau}^{\prime}$ and the “machine load” $L_{o_\tau} $ are machine-level quantities (how long the assigned machine will remain busy and how heavily it is loaded) that are pre-aggregated and attached to each operation node. This pools information that is specific to machines into operation nodes, rather than representing machine nodes. As a result, the network processes only operation nodes and does not explicitly model machine nodes or operation–machine edges, which makes a direct extension to FJSP with flexible machine choices non-trivial.
>
> Second, some features (such as Features 3, which is the sum of remaining processing times along the job route) rely on the linear job structure of JSSP. Such heuristic design can not directly generalize to more general precedence graphs where operations can have parallel predecessors.
>
> In contrast, our approach starts from a heterogeneous graph perspective, in which JSSP and FFSP naturally appear as special cases of FJSP. This perspective allows us to design a unified yet simplified state representation and a Transformer-based policy network that explicitly model both operation and machine nodes as well as O2O/O2M edges, while deliberately avoiding trapped into problem-specific designs. As a result, the same model can be directly applied to FJSP, JSSP, and FFSP without modifying the state definition or network structure, achieving a genuine “one model for multiple variants” setting. Moreover, our formulation is readily extensible to scheduling problems with more general precedence structures (e.g., assembly job-shop with parallel dependencies), where the problem-specific structure is captured only through the input graph and masks, without requiring additional hand-crafted features or architectural changes.
>
> [AR4]. Lee J H, Kim H J. Graph-based imitation learning for real-time job shop dispatcher. IEEE Transactions on Automation Science and Engineering, 2024.

---

> ### Author Response · Authors · 2025-12-01
> **Response (2/3)**
>
> > **W3**: JSP comparisons are incomplete and relatively weak. While FJSP results include several recent methods, JSP needs stronger baselines. For example: – Park, J., Bakhtiyar, S., & Park, J. (2021). ScheduleNet. arXiv:2106.03051. – Chen, R., Li, W., & Yang, H. (2022). TII 19(2):1322–1331. – Iklassov, Z. et al. (2023). IJCAI, pp. 5350–5358. – Lee, J. H., & Kim, H. J. (2025). Graph-based imitation learning for real-time job shop dispatcher. IEEE T-ASE.
>
> We thank the reviewer for referring these recent learning based methods for JSP to us. Our main focus in this work is FJSP, and JSSP/FFSP are included primarily as additional testbeds to demonstrate that our unified framework can handle multiple scheduling variants. For this reason, in Table 3 we chose widely used JSSP[AR5, AR6] baselines under the Taillard/DMU offline setting, rather than aiming at an exhaustive comparison with all recent JSP-specific methods.
>
> Regarding DGERD[AR7], we ran an additional comparison following their evaluation protocol. Using our ReSched model trained on synthetic 10×10 instances, we evaluated all sizes of the Taillard benchmark and, as in their paper, reported the first two instances for each size. For both methods, the optimality gaps are computed with respect to the best-known upper bounds (UB) reported in the literature. The results in Table RT3 show that ReSched consistently achieves smaller makespans and lower optimality gaps than DGERD on all 16 instances (average gap 30.6% vs. 19.9%). This indicates that, even under their evaluation setting, our approach is competitive with this recent JSP-specific method.
>
> **Table RT3. Comparison with DGERD on the Taillard JSP benchmark**
> |Instance|Size|DGERD Makespan|DGERD Gap|ReSched Makespan|ReSched Gap|UB|
> |-|-|-|-|-|-|-|
> |ta01|15×15|1711|39.0|**1515**|**23.1**|1231|
> |ta02|15×15|1639|31.8|**1415**|**13.7**|1244|
> |ta11|20×15|1833|35.1|**1658**|**22.2**|1357|
> |ta12|20×15|1765|29.1|**1711**|**25.2**|1367|
> |ta21|20×20|2146|30.7|**1930**|**17.5**|1642|
> |ta22|20×20|2016|26.0|**1784**|**11.5**|1600|
> |ta31|30×15|2383|35.1|**2266**|**28.5**|1764|
> |ta32|30×15|2459|37.8|**2163**|**21.2**|1784|
> |ta41|30×20|2541|26.7|**2522**|**25.7**|2006|
> |ta42|30×20|2762|42.4|**2395**|**23.5**|1939|
> |ta51|50×15|3763|36.3|**3417**|**23.8**|2760|
> |ta52|50×15|3511|27.4|**3372**|**22.4**|2756|
> |ta61|50×20|3633|26.7|**3367**|**17.4**|2868|
> |ta62|50×20|3712|29.4|**3418**|**19.1**|2869|
> |ta71|100×20|6321|15.7|**6074**|**11.2**|5464|
> |ta72|100×20|6232|20.3|**5783**|**11.6**|5181|
> |Avg.|-|3026.7|30.6|**2799.4**|**19.9**|2364.5|
>
> While our focus is FJSP, in the final version, we still plan to additionally include ScheduleNet[AR8], the curriculum-learning based method of Iklassov et al.[AR9], and the graph-based dispatcher of Lee and Kim[AR4] in our JSSP discussion and comparisons where appropriate, so that the positioning of our DRL baselines for JSP is more complete.
>
> [AR4]. Lee J H, Kim H J. Graph-based imitation learning for real-time job shop dispatcher. IEEE Transactions on Automation Science and Engineering, 2024.
>
> [AR5]. Zhang C, Song W, Cao Z, et al. Learning to dispatch for job shop scheduling via deep reinforcement learning. Advances in neural information processing systems, 2020.
>
> [AR6]. Park J, Chun J, Kim S H, et al. Learning to schedule job-shop problems: representation and policy learning using graph neural network and reinforcement learning. International journal of production research, 2021.
>
> [AR7]. Chen R, Li W, Yang H. A deep reinforcement learning framework based on an attention mechanism and disjunctive graph embedding for the job-shop scheduling problem. IEEE Transactions on Industrial Informatics, 2022.
>
> [AR8]. Park J, Bakhtiyar S, Park J. Schedulenet: Learn to solve multi-agent scheduling problems with reinforcement learning. arXiv, 2021.
>
> [AR9]. Iklassov Z, Medvedev D, De Retana R S O, et al. On the Study of Curriculum Learning for Inferring Dispatching Policies on the Job Shop Scheduling. IJCAI. 2023.

---

> ### Author Response · Authors · 2025-12-01
> **Response (3/3)**
>
> > **Q2**: You excel on FJSP but are less competitive on recent JSP baselines. What structural or distributional differences make JSP harder for your method?
>
> We agree with the reviewer that there is a clear trend: as the instance distribution moves closer to a pure JSSP (one feasible machine per operation), the relative advantage of FJSP-oriented DRL methods becomes smaller. This phenomenon is not specific to ReSched; it also appears on existing FJSP baselines. For example, on the Hurink FJSP benchmarks, edata has on average only 1.15 feasible machines per operation (very close to JSSP), whereas rdata has 2 and vdata has about \(m/2\) machines per operation. As shown in Table below, all three FJSP methods (HGNN, DANIEL, and ReSched) perform worst on edata and improve as the flexibility increases from edata → rdata → vdata.
>
> **Table RT4. Perfprmance degradation analysis for RL approaches. Values are optimality gaps（%） to the optimal solutions.**
> |Dataset|Avg. number of machine per op|HGNN 10×5|DANIEL 10×5|ReSched 10×5|
> |-|-|-|-|-|
> |Hurink(edata)|1.15|15.53|16.33|15.48|
> |Hurink(rdata)|2|11.15|11.42|10.18|
> |Hurink(vdata)|m/2|4.25|3.28|3.48|
>
>
> Beyond the empirical trend observed on the Hurink dataset, we would like to highlight a structural reason for this behavior. Methods for the Flexible JSP (FJSP), including our own, are explicitly designed to model the interactions between operations and machines. Our architecture, for instance, incorporates a dedicated O2M branch and machine nodes, whose primary function is to resolve assignments among multiple feasible machines and balance their loads.
>
> In strict JSP instances, however, this flexibility is absent, as each operation is assigned to a single feasible machine. Consequently, the O2M decision structure becomes less critical, and a portion of the model's dedicated capacity is consequently less informative. The challenge in these instances shifts predominantly to resolving fine-grained sequencing conflicts along predetermined machine routes.
>
> In contrast, many recent JSP-tailored methods simplify the representation by pooling machine-related information into aggregated operation features, effectively modeling the problem on an operation-only graph[AR4, AR5, AR6, AR7, AR8]. This abstraction can enable the model to concentrate more directly on resolving sequencing conflicts within a fixed disjunctive graph, which may yield stronger performance on classic JSP benchmarks.
>
> However, we posit that in practical manufacturing settings, an operation is intrinsically linked to its machine resources, as embodied by attributes like machine-specific processing times. This is why our framework intentionally maintains distinct operation and machine nodes with explicit O2O and O2M edges. This architectural choice necessarily trades off some degree of short-term, JSP-specific optimality to achieve a more generalizable, faithful, and extensible model. It is specifically designed to accommodate FJSP and other complex variants where machine flexibility and richer precedence structures are fundamental to the problem.
>
> [AR4]. Lee J H, Kim H J. Graph-based imitation learning for real-time job shop dispatcher. IEEE Transactions on Automation Science and Engineering, 2024.
>
> [AR5]. Zhang C, Song W, Cao Z, et al. Learning to dispatch for job shop scheduling via deep reinforcement learning. Advances in neural information processing systems, 2020.
>
> [AR6]. Park J, Chun J, Kim S H, et al. Learning to schedule job-shop problems: representation and policy learning using graph neural network and reinforcement learning. International journal of production research, 2021.
>
> [AR7]. Chen R, Li W, Yang H. A deep reinforcement learning framework based on an attention mechanism and disjunctive graph embedding for the job-shop scheduling problem. IEEE Transactions on Industrial Informatics, 2022.
>
> [AR8]. Park J, Bakhtiyar S, Park J. Schedulenet: Learn to solve multi-agent scheduling problems with reinforcement learning. arXiv, 2021.

---

### Official Review · Reviewer_1Xow · 2025-11-01

**Soundness:** 3
**Presentation:** 2
**Contribution:** 3
**Rating:** 6
**Confidence:** 4

**Summary:**

The paper proposes RESCHED, a deep reinforcement learning framework for solving scheduling problems. It proposes a simplified state representation and Transformers to solve Job Shop Scheduling Problems. The experiments show better performance than DANIEL, and RESCHED is even surpassing OR-TOOLS on 40x10 instances. The architecture used is a decoder only architecture and the reinforcement learning algorithm is REINFORCE. It learns a policy that is then used either greedily or with sampling.

**Strengths:**

The paper has good results on large instances.
A new state representation and architecture are proposed.

**Weaknesses:**

The writing is not polished (Experments, Dateset, ...)
The training times of the different DRL approaches are not compared.
It would be better to also have OR-TOOLS on Taillard benchmark.

**Questions:**

Can you detail the sampling algorithm used in Table 1?

---

> ### Author Response · Authors · 2025-12-01
> **Response to Weaknesses (part1/2)**
>
> > **W1**: The writing is not polished (Experments, Dateset, ...). The **training times** of the different DRL approaches are not compared. It would be better to also have **OR-TOOLS on Taillard benchmark**.
>
> We appreciate the reviewer’s time on reviewing our paper and the valuable comments.
>
> **Regarding Typos:** Thanks for pointing out those typos. We have corrected “Experments” and “Dateset”, along with other minor grammatical issues in the revised paper.
>
> **Regarding Training Cost:** We agree with the reviewer that training cost is a critically important practical factor. Our initial implementation utilized a REINFORCE-style policy gradient, which indeed makes a direct comparison with PPO-based methods less straightforward. The REINFORCE-based ReSched requires approximately 30 minutes to train on the SD1-10×5 instances. In response to this comment and suggestions from other reviewers, for fair comparison, we additionally implemented a PPO-based version of our approach. Under the same training budget, the PPO version trains in about 20 minutes for the SD1-10×5 instances.This reduction is mainly because PPO reuses each collected batch for multiple policy/critic updates, which reduces the number of environment interactions required for the same number of gradient updates. On the same hardware and with the same computational configuration, HGNN [AR2] and DANIEL [AR3] require about 21 minutes and 18 minutes, respectively. These results demonstrate that our method achieves comparable training efficiency as other RL approaches.
>
> As the reviewer suggested, we also have undertaken a comprehensive benchmarking of our PPO-based ReSched against the state-of-the-art FJSP method, DANIEL [AR3], evaluating performance on both in-distribution and generalization scenarios. This comparative analysis reveals that the PPO implementation offers a dual advantage: (i) it achieves superior overall performance on in-distribution test instances under an equivalent training budget, and (ii) it exhibits robust generalization, effectively scaling from smaller training sizes to larger, unseen instances. The results are presented in Table.RT1 below.
>
> **Table RT1. REINFORCE based ReSched vs.  PPO based ReSched in optimality gap (%). The value in bold represents best performance.**
> |In-distribution|DANIEL|ReSched-REINFORCE|ReSched-PPO|
> |-|-|-|-|
> |SD1-10x05|**10.87**|14.61|11.20|
> |SD1-20x05|**5.03**|8.51|5.84|
> |SD1-15x10|12.42|12.91|**9.41**|
> |SD1-20x10|**1.31**|6.97|3.50|
> |SD2-10x05|25.68|19.06|**15.77**|
> |SD2-20x05|11.52|10.76|**9.21**|
> |SD2-15x10|57.16|**24.03**|25.02|
> |SD2-20x10|31.58|**17.91**|19.38|
> |Avg.|19.45|14.35|**12.42**|
> |**Out-of-distribution**|**DANIEL**|**ReSched-REINFORCE**|**ReSched-PPO**|
> |SD1-30x10|5.10|9.26|**3.21**|
> |SD1-40x10|3.65|8.04|**2.19**|
> |SD2-30x10|14.85|37.76|**8.97**|
> |SD2-40x10|0.52|23.25|**-3.51**|
> |Avg.|6.03|19.58|**2.72**|
>
> We would like to take this opportunity to clarify that for deep reinforcement learning, wall-clock training time is highly dependent on environment interaction, including simulation specifics and implementation efficiency. Consequently, such training times reported across different studies are often not directly comparable. The principal advantage of our DRL approach, lies in its fast inference capability， particularly due to its construction-based nature. As demonstrated in Figure 3 of the manuscript, our method achieves inference times comparable to both HGNN and DANIEL, while simultaneously delivering substantially superior performance, thereby highlighting its significant practical value.
>
> [AR1]. Da Col, Giacomo, and Erich C. Teppan. Industrial size job shop scheduling tackled by present day CP solvers. International Conference on Principles and Practice of Constraint Programming, 2019.
>
> [AR2]. Song W, Chen X, Li Q, et al. Flexible job-shop scheduling via graph neural network and deep reinforcement learning. IEEE Transactions on Industrial Informatics, 2022.
>
> [AR3]. R. Wang, G. Wang, J. Sun, F. Deng and J. Chen, Flexible Job Shop Scheduling via Dual Attention Network-Based Reinforcement Learning. IEEE Transactions on Neural Networks and Learning Systems, 2024

---

> ### Author Response · Authors · 2025-12-01
> **Response to Weaknesses (part2/2)**
>
> > **W1**: The writing is not polished (Experments, Dateset, ...). The **training times** of the different DRL approaches are not compared. It would be better to also have **OR-TOOLS on Taillard benchmark**.
>
> **Regarding ORTools on Taillard Benckmark:** Thanks for the reviewers' suggestion. In the revised manuscript (Sec. 5, Table 3), we have added the OR-Tools results on the Taillard benchmark for reference. Specifically, we use the CP-SAT[AR1] solver in OR-Tools with a 3600-second time limit per instance and report the best solution found within this limit.
>
> **Table RT2. OR-Tools results on Taillard benchmark. Values are optimality gaps（%） to the optimal solutions.**
> |Size|OR-Tools|SPT|MWKR|FDD/MWKR|MOPNR|L2D|RL-GNN|ReSched 10×10|UB|
> |-|-|-|-|-|-|-|-|-|-|
> |15×15|0.1|54.8|56.7|47.1|45.0|26.0|20.1|**15.7**|1233.9|
> |20×15|0.2|65.2|60.7|50.6|47.7|30.0|24.9|**19.7**|1361.3|
> |20×20|0.7|64.2|55.7|47.6|42.8|31.6|29.2|**16.3**|1617.1|
> |30×15|2.1|61.6|52.6|45.0|45.6|33.0|24.7|**21.5**|1771.2|
> |30×20|2.8|66.0|63.9|56.3|48.2|33.6|32.0|**22.5**|1919.4|
> |50×15|0.0|51.4|40.9|34.8|30.1|22.4|15.9|**16.1**|2783.8|
> |50×20|2.8|59.5|53.9|41.5|37.9|26.5|21.3|**15.6**|2834.4|
> |100×20|3.9|41.0|32.9|23.4|20.2|13.6|9.2|**9.6**|5369.6|
>
> [AR1]. Da Col, Giacomo, and Erich C. Teppan. Industrial size job shop scheduling tackled by present day CP solvers. International Conference on Principles and Practice of Constraint Programming, 2019.

---

> ### Author Response · Authors · 2025-12-01
> **Details of the Sampling-Based Evaluation Protocol**
>
> > **Q1**: Can you detail the sampling algorithm used in Table 1?
>
> We are grateful to the reviewer for raising this point, which has allowed us to clarify the methodology. In Table 1, the reported "sampling" performance is obtained by repeatedly sampling solutions from the stochastic policy and selecting the best outcome. Specifically, aligning with the methodologies of HGNN [AR2] and DANIEL [AR3], we execute 100 independent decoding trajectories for each test instance. Within each trajectory, the next operation–machine pair is sampled directly from the categorical distribution output by the policy network, without employing any additional search or local improvement techniques. Upon termination of all trajectories, the solution achieving the smallest makespan is selected for reporting. We have incorporated a clarification of this procedure in the revised manuscript (Section 5).
>
> [AR2]. Song W, Chen X, Li Q, et al. Flexible job-shop scheduling via graph neural network and deep reinforcement learning. IEEE Transactions on Industrial Informatics, 2022.
> [AR3]. R. Wang, G. Wang, J. Sun, F. Deng and J. Chen, Flexible Job Shop Scheduling via Dual Attention Network-Based Reinforcement Learning. IEEE Transactions on Neural Networks and Learning Systems, 2024

---

### Meta-Review · Area_Chair_vjiT · 2026-01-07

**Summary:**

The reviewers raised concerns spanning four major categories:
* Algorithm and Training Fairness: Reviewer WVwe challenged the use of high-variance REINFORCE over PPO, questioned training data fairness (claiming 5000× more data than baselines), and criticized the lack of multi-seed robustness tests. Reviewer 54We also questioned the REINFORCE choice and requested matched-budget comparisons.
* Theoretical Rigor: Reviewer WVwe and cunA both flagged that Proposition 1's "minimal MDP" claim lacked formal proof, with WVwe noting Corollary 1 depended on an unproven proposition.
* Experimental Completeness: Multiple reviewers identified gaps: missing runtime comparisons (54We, 1Xow), no OR-Tools Taillard benchmark (1Xow), unclear sampling algorithm (1Xow), ambiguous JSP evaluation strategy (WVwe), incomplete JSP baselines (cunA), and no comparison with feature-reduced DANIEL (54We).
* Presentation and Scope: Reviewers cited unclear figures (54We), typos (1Xow), marginal significance of contributions (WVwe), missing metaheuristic comparisons (54We), and limited discussion on extending to real-world constraints (54We).

**Reviewer Concerns:**

Concerns addressed:
* Training algorithm: Implemented PPO version, showing it converges faster and improves performance; demonstrated robustness across 4 seeds with <1% variance.
* Training budget: Clarified the budget gap is ~25× (not 5000×) and provided extensive matched-budget experiments under both small (80K trajectories) and large (2M trajectories) settings.
* Theoretical proof: Added formal proof of Proposition 1 in Appendix D.
* Runtime analysis: Reported training times (PPO: 20min, REINFORCE: 30min vs. DANIEL: 18min, HGNN: 21min) and inference times across all benchmarks.
* OR-Tools benchmark: Added Table RT2 with CP-SAT results on Taillard JSP.
* Sampling algorithm: Clarified 100 independent trajectories sampled from policy, selecting best makespan.
* JSP evaluation: Explicitly confirmed greedy strategy used for Table 3.
* Feature-reduced DANIEL: Comprehensive Table RT6 shows ReSched still outperforms DANIEL-Del across in-distribution, OOD, and open benchmarks.
* Related work: Added Lee & Kim (2024) discussion and comparison, plus DGERD JSP baseline.
* JSP performance gap: Explained structural reasons (FJSP methods' O2M modeling less critical in pure JSP).
* Figures: Modified Fig. 1a and enhanced Fig. 2 captions.
* Machine embeddings: Clarified two-level removal process in ablation.
* Reward function: Tested three variants, showing lower-bound reward best for generalization.
* RoPE encoding: Provided rationale and comparison showing similar performance to index-based.
* Metaheuristics: Added 2SGA comparison and runtime-quality tradeoff discussion.
* Extension: Provided mobile edge computing example for broader applicability.
* Hyperparameters: Tested discount factor $\gamma$=1.0 vs 0.99, showing insensitivity.

As for Reviewer WVwe's concern about marginality remains subjective despite authors' defense of foundation-style approach importance, I think the new state representation and network architecture is valuable enough for the work. Thus I think most of the concerns have been well addressed.

**Reviewer Scores:**

WVwe's concerns were the most severe and numerous, but also the most directly addressed. The rebuttal provides:
* PPO implementation showing faster convergence and better performance
* Multi-seed experiments (4 seeds) demonstrating <1% variance, directly refuting stability concerns
* Budget clarification (25× not 5000×) and extensive matched-budget experiments under both small (80K) and large (2M) settings
* Formal proof added to Appendix D
* Feature-reduced DANIEL comparison showing ReSched still significantly outperforms it
However, WVwe's fundamental skepticism about the "marginality" of the contribution (state representation + architecture) is philosophical and harder to resolve. While the unified framework and minimal design are convincingly defended, a reviewer who initially scored 2 likely harbors deeper reservations about novelty that exhaustive experiments may only partially ameliorate. So I think the reviewer may raise score from 2 to 4.

As for other reviewers, I think the scores will be maintained at 6 or raised to 8. Thus my overall assesement to the paper is above the ICLR threshold.

---

### Decision · Program_Chairs · 2026-01-26

Accept (Poster)